# TOPOLOGY-AWARE EMBEDDING MEMORY FOR LEARNING ON EXPANDING GRAPHS

## ABSTRACT

Memory replay based techniques have shown great success for continual learning with incrementally accumulated Euclidean data. Directly applying them to continually expanding graphs, however, leads to the potential memory explosion problem due to the need to buffer representative nodes and their associated topological neighborhood structures. To this end, we systematically analyze the key challenges in the memory explosion problem, and present a general framework, *i.e.*, Parameter Decoupled Graph Neural Networks (PDGNNs) with Topology-aware Embedding Memory (TEM), to tackle this issue. The proposed framework not only reduces the memory space complexity from $\mathcal{O}(nd^L)$ to $\mathcal{O}(n)$ [1], but also fully utilizes the topological information for memory replay. Specifically, PDGNNs decouple trainable parameters from the computation ego-subgraph via *Topology-aware Embeddings* (TEs), which compress ego-subgraphs into compact vectors (*i.e.*, TEs) to reduce the memory consumption. Based on this framework, we discover a unique *pseudo-training effect* in continual learning on expanding graphs and this effect motivates us to develop a novel *coverage maximization sampling* strategy that can enhance the performance with a tight memory budget. Thorough empirical studies demonstrate that, by tackling the memory explosion problem and incorporating topological information into memory replay, PDGNNs with TEM significantly outperform state-of-the-art techniques, especially in the challenging class-incremental setting.

## 1 INTRODUCTION

Traditional graph learning works typically assume the graph to be static. However, real-world graphs often expand constantly with emerging new types of nodes and their associated edges. Accordingly, models trained incrementally on the new node types may experience catastrophic forgetting (severe performance degradation) on the old ones. Targeting this challenge, continual learning on expanding graphs (Liu et al., 2021; Zhou & Cao, 2021; Zhang et al., 2022c) attracts increasingly more attention recently. It exhibits enormous value in various practical applications, especially in the case where graphs are relatively large, and retraining a new model over the entire graph is computationally infeasible. For instance, in a social network, a community detection model has to keep adapting its parameters based on nodes from newly emerged communities; in a citation network, a document classifier needs to continuously update its parameters to distinguish the documents of newly emerged research fields.

Memory replay (Rebuffi et al., 2017; Lopez-Paz & Ranzato, 2017; Aljundi et al., 2019; Shin et al., 2017), which stores representative samples in a buffer to retrain the model and maintain its performance over existing tasks, exhibits great success in preventing catastrophic forgetting for various continual learning tasks, *e.g.*, computer vision and reinforcement learning (Kirkpatrick et al., 2017; Li & Hoiem, 2017; Aljundi et al., 2018; Rusu et al., 2016). Directly applying memory replay to graph data with the popular message passing neural networks (MPNNs, the general framework for most GNNs) (Gilmer et al., 2017; Kipf & Welling, 2017; Veličković et al., 2017), however, could give rise to the memory explosion problem because the necessity to consider the explicit topological information of target nodes. Specifically, due to the message passing over the topological connections in graphs, retraining an $L$-layer GNN (Figure 1, left) with $n$ buffered nodes would require storing $\mathcal{O}(nd^L)$ nodes (Chiang et al., 2019; Chen et al., 2017) (the number of edges is not counted yet) in the buffer, where $d$ is the average node degree. Take the Reddit dataset (Hamilton et al., 2017) as an example, its average node degree is 492, and the buffer size will easily be intractable even with a 2-layer GNN. To resolve this issue, Experience Replay based GNN (ER-GNN) (Zhou & Cao, 2021) stores representative input nodes (*i.e.*, node attributes) in the buffer but completely ignores the

---

[1] $n$: memory budget, $d$: average node degree, $L$: the radius of the GNN receptive field

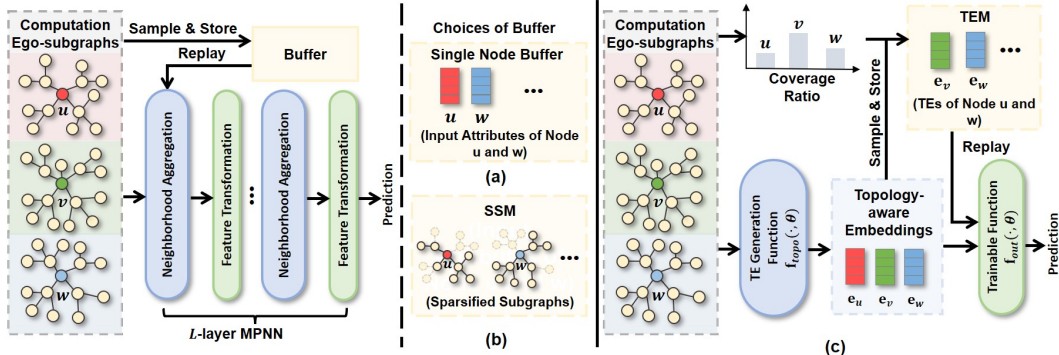

Figure 1: (a) ER-GNN (Zhou & Cao, 2021) that stores the *input attributes* of individual nodes. (b) Sparsified Subgraph Memory (SSM) (Zhang et al., 2022d) that stores sparsified computation ego-subgraphs. (c) Our PDGNNs with TEM. The incoming computation ego-subgraphs are embedded as TEs and then fed into the trainable function. The stored TEs are sampled based on their coverage ratio (Section 3.6).

topological information (Figure 1 a). Feature graph network (FGN) (Wang et al., 2020a) implicitly encodes node proximity with the inner products between the features of the target node and its neighbors. However, the explicit topological connections are abandoned and message passing is no longer feasible on the graph. Sparsified Subgraph Memory (SSM) (Zhang et al., 2022d) sparsifies the computation ego-subgraphs for tractable memory consumption, which still partially sacrifices topological information, especially when the computation ego-subgraphs are large and a majority of nodes/edges is removed after sparsification (Figure 1 b).

To this end, we present a general framework of Parameter Decoupled Graph Neural Networks (PDGNNs) with Topology-aware Embedding Memory (TEM) to perform continual learning on expanding graphs (Figure 1 c). First, with systematic analysis, we demonstrate that the necessity to store the complete computation ego-subgraphs for retraining MPNNs arises from the entanglement between the trainable parameters and the individual nodes/edges (Section 3.2). Targeting this problem, we design the Parameter Decoupled Graph Neural Networks (PDGNNs) framework, which decouples the trainable parameters from individual nodes/edges. The PDGNNs framework enables us to develop a novel concept, *Topology-aware Embedding* (TE), which is a vector with a fixed size but contains all necessary information for retraining the trainable parameters of PDGNNs. Such TEs are desired surrogates of computation ego-subgraphs to facilitate memory replay. After learning each task, a subset of TEs is selected with a certain sampling strategy and stored in the *Topology-aware Embedding Memory* (TEM). Because the size of a TE is fixed, the memory space complexity of a buffer (with size $n$) can be dramatically reduced from $\mathcal{O}(nd^L)$ to $\mathcal{O}(n)$. Moreover, different from traditional continual learning on Euclidean data without explicit topological connections (*e.g.*, images), we theoretically discover that replaying the TE of one single node incurs a *pseudo-training effect* on the neighboring nodes, which could also alleviate the forgetting problem for the other nodes within the same computation ego-subgraph. This unique phenomenon in continual learning on expanding graphs takes place due to the neighborhood aggregation in GNNs. The pseudo-training effect suggests that TEs corresponding to larger computation ego-subgraphs (quantitatively measured by coverage ratio) are more beneficial to continual learning performance. Based on the theoretical finding, we develop a novel *coverage maximization sampling* strategy, which enlarges the coverage ratio of the selected TEs and empirically enhances the performance for a tight memory budget. In our experiments, we adopt both the class-incremental (class-IL) continual learning scenario (Rebuffi et al., 2017; Zhang et al., 2022d;b) and the task-incremental (task-IL) scenario (Liu et al., 2021; Zhou & Cao, 2021), to evaluate the effectiveness of PDGNNs-TEM. Thorough empirical studies demonstrate that PDGNNs with TEM outperform the state-of-the-art techniques in both class-IL and task-IL scenarios.

## 2 RELATED WORKS

### 2.1 CONTINUAL LEARNING & CONTINUAL LEARNING ON EXPANDING GRAPHS

To alleviate the catastrophic forgetting problem, existing approaches can be categorized into regularization, memory replay, and parameter isolation based methods. Regularization based methods aim to prevent drastic modification to parameters that are important for previous tasks (Farajtabar et al., 2020; Kirkpatrick et al., 2017; Li & Hoiem, 2017; Aljundi et al., 2018; Hayes & Kanan, 2020; Rakaraddi et al., 2022; Sun et al., 2022; Qi et al.). Parameter isolation methods adaptively allocate new parameters for the new tasks to protect the ones for the previous tasks (Wortsman et al., 2020; Wu et al., 2019b; Yoon et al., 2020; 2017; Rusu et al., 2016). Memory replay based methods store and

replay representative data from previous tasks when learning new tasks (Caccia et al., 2020; Chrysakis & Moens, 2020; Rebuffi et al., 2017; Lopez-Paz & Ranzato, 2017; Aljundi et al., 2019; Shin et al., 2017). Recently, continual learning on expanding graphs attracts increasingly more attention due to its practical importance (Das & Isufi, 2022a;c;b; Zhou & Cao, 2021; Liu et al., 2021; Wang et al., 2020b; Xu et al., 2020; Daruna et al., 2021; Carta et al., 2021; Zhang et al., 2022d;c;b; Febrinanto et al., 2023; Yuan et al., 2023; Cai et al., 2022; Wei et al., 2022; Wang et al., 2022; Ahrabian et al., 2021; Su & Wu, 2023; Kim et al., 2022; Daruna et al., 2021; Hoffmann et al., 2023; Galke et al., 2023; Li et al., 2023; Cui et al., 2023). Existing methods include regularization ones like topology-aware weight preserving (TWP) (Liu et al., 2021) that preserves crucial topologies, parameter isolation methods like HPNs (Zhang et al., 2022c) that select different parameters for different tasks, and memory replay methods like ER-GNN (Zhou & Cao, 2021) and Sparsified Subgraph Memory (SSM) (Zhang et al., 2022d) that store representative nodes or sparsified computation ego-subgraphs. Our work is also memory based and its key advantage is the capability to preserve complete topological information with reduced space complexity, which shows significant superiority in class-IL setting (Section 4.4). Finally, it is worth highlighting the difference between continual learning on expanding graphs and some relevant research areas. First, dynamic graph learning (Galke et al., 2021; Wang et al., 2020c; Han et al., 2020; Yu et al., 2018; Nguyen et al., 2018; Zhou et al., 2018; Ma et al., 2020; Feng et al., 2020) focuses on the temporal dynamics with all previous data being accessible. In contrast, continual learning on expanding graphs aims to alleviate forgetting, therefore the previous data is inaccessible. Second, few-shot graph learning (Zhou et al., 2019; Guo et al., 2021; Yao et al., 2020; Tan et al., 2022) targets fast adaptation to new tasks. In training, few-shot learning models can access all previous tasks (unavailable in continual learning). In testing, few-shot learning models need to be fine-tuned on the test classes, while the continual learning models are tested on existing tasks without fine-tuning.

## 2.2 Graph Neural Networks & Reservoir Computing

Graph Neural Networks (GNNs) are deep learning models designed to generate representations for graph data, which typically interleave the neighborhood aggregation and node feature transformation to extract the topological features (Kipf & Welling, 2017; Gilmer et al., 2017; Veličković et al., 2017; Xu et al., 2018; Chen et al., 2018; Hamilton et al., 2017; Yan et al., 2022; Yang et al., 2023; Zhang et al., 2022a). GNNs without interleaving the neighborhood aggregation and node feature transformation have been developed to reduce the computation complexity and increase the scalability (Zeng et al., 2021; Chen et al., 2020; 2019; Nt & Maehara, 2019; Frasca et al., 2020; Fey et al., 2021; Cong et al., 2020; Dong et al., 2021). For example, Simple Graph Convolution (SGC) (Wu et al., 2019a) removes the non-linear activation from GCN (Kipf & Welling, 2017) and only keeps one neighborhood aggregation and one node transformation layer. Approximate Personalized Propagation of Neural Predictions (APPNP) (Klicpera et al., 2018) first performs node transformation and then conducts multiple neighborhood aggregations in one layer. Motivated by these works, the PDGNNs framework in this paper is specially designed to decouple the neighborhood aggregation with trainable parameters, and derive the topology-aware embeddings (TEs) to reduce the memory space complexity and facilitate continual learning on expanding graphs. Besides, PDGNNs are also related to reservoir computing (Gallicchio & Micheli, 2020; 2010), which can embed the input data (*e.g.* graphs) via a fixed non-linear system. The reservoir computing modules can be adopted in PDGNNs as the TE generation function (Equation 4), and the corresponding experiments are included in Appendix B.2.

## 3 Parameter Decoupled GNNs with Topology-aware Embedding Memory

In this section, we first introduce the notations, and then explain the technical challenge of applying memory replay techniques to GNNs. Targeting the challenge, we introduce PDGNNs with *Topology-aware Embedding Memory* (TEM). Finally, inspired by theoretical findings of the *pseduo-training effect*, we develop the coverage maximization sampling to enhance the performance when the memory budget is tight, which has shown its effectiveness in our empirical study. All detailed proofs are provided in Appendix A.

### 3.1 Preliminaries

Continual learning on expanding graphs is formulated as learning node representations on a sequence of subgraphs (tasks): $\mathcal{S} = \{\mathcal{G}_1, \mathcal{G}_2, ..., \mathcal{G}_T\}$. Each $\mathcal{G}_\tau$ (*i.e.*, $\tau$-th task) contains several new categories of nodes in the overall graph, and is associated with a node set $\mathbb{V}_\tau$ and an edge set $\mathbb{E}_\tau$, which

is represented as the adjacency matrix $\mathbf{A}_\tau \in \mathbb{R}^{|\mathbb{V}_\tau| \times |\mathbb{V}_\tau|}$. The degree of a node $d$ refers to the number of edges connected to it. In practice, $\mathbf{A}_\tau$ is often normalized as $\hat{\mathbf{A}}_\tau = \mathbf{D}_\tau^{-\frac{1}{2}} \mathbf{A}_\tau \mathbf{D}_\tau^{-\frac{1}{2}}$, where $\mathbf{D}_\tau \in \mathbb{R}^{|\mathbb{V}_\tau| \times |\mathbb{V}_\tau|}$ is the degree matrix. Each node $v \in \mathbb{V}_\tau$ has a feature vector $\mathbf{x}_v \in \mathbb{R}^b$. In classification tasks, each node $v$ has a label $\mathbf{y}_v \in \{0,1\}^C$, where $C$ is the total number of classes. When generating the representation for a target node $v$, a L-layer GNN typically takes a computation ego-subgraph $\mathcal{G}_{\tau,v}^{sub}$, containing the $L$-hop neighbors of $v$ (*i.e.* $\mathcal{N}^L(v)$), as the input. For simplicity, $\mathcal{G}_v^{sub}$ is used in the following, without the graph index.

## 3.2 MEMORY REPLAY MEETS GNNs

In traditional continual learning, a model $f(\cdot; \boldsymbol{\theta})$ parameterized by $\boldsymbol{\theta}$ is sequentially trained on $T$ tasks. Each task $\tau$ ($\tau \in \{1, ..., T\}$) corresponds to a dataset $\mathbb{D}_\tau = \{(\mathbf{x}_i, \mathbf{y}_i)_{i=1}^{n_\tau}\}$. To avoid forgetting, memory replay based methods store representative data from the old tasks in a buffer $\mathcal{B}$. When learning new tasks. A common approach to utilize $\mathcal{B}$ is through an auxiliary loss:

$$\mathcal{L} = \underbrace{\sum_{\mathbf{x}_i \in \mathbb{D}_\tau} l(f(\mathbf{x}_i; \boldsymbol{\theta}), \mathbf{y}_i)}_{\mathcal{L}_\tau: \text{ loss of the current task}} + \lambda \underbrace{\sum_{\mathbf{x}_j \in \mathcal{B}} l(f(\mathbf{x}_j; \boldsymbol{\theta}), \mathbf{y}_j)}_{\mathcal{L}_{aux}: \text{ auxiliary loss}}, \tag{1}$$

where $l(\cdot, \cdot)$ denotes the loss function, and $\lambda \geq 0$ balances the contribution of the old data. Instead of directly minimizing $\mathcal{L}_{aux}$, the buffer $\mathcal{B}$ may also be used in other ways to prevent forgetting (Lopez-Paz & Ranzato, 2017; Rebuffi et al., 2017). In these applications, the space complexity of a buffer containing $n$ examples is $\mathcal{O}(n)$.

However, to capture the topological information, GNNs obtain the representation of a node $v$ based on a computation ego-subgraph surrounding $v$. We exemplify it with the popular MPNN framework (Gilmer et al., 2017), which updates the hidden node representations at the $l + 1$-th layer as:

$$\mathbf{m}_v^{l+1} = \sum_{w \in \mathcal{N}^1(v)} M_l(\mathbf{h}_v^l, \mathbf{h}_w^l, \mathbf{x}_{v,w}^e; \boldsymbol{\theta}_l^M), \qquad \mathbf{h}_v^{l+1} = U_l(\mathbf{h}_v^l, \mathbf{m}_v^{l+1}; \boldsymbol{\theta}_l^U), \tag{2}$$

where $\mathbf{h}_v^l$, $\mathbf{h}_w^l$ are hidden representations of nodes at layer $l$, $\mathbf{x}_{v,w}^e$ is the edge feature, $M_l(\cdot, \cdot, \cdot; \boldsymbol{\theta}_l^M)$ is the message function to integrate neighborhood information, and $U_l(\cdot, \cdot; \boldsymbol{\theta}_l^U)$ updates $\mathbf{m}_v^{l+1}$ into $\mathbf{h}_v^l$ ($\mathbf{h}_v^0$ is the input features). In a $L$-layer MPNN, the representation of a node $v$ can be simplified as,

$$\mathbf{h}_v^L = \text{MPNN}(\mathbf{x}_v, \mathcal{G}_v^{sub}; \boldsymbol{\Theta}), \tag{3}$$

where $\mathcal{G}_v^{sub}$ contains the $L$-hop neighbors ($\mathcal{N}^L(v)$), $\text{MPNN}(\cdot, \cdot; \boldsymbol{\Theta})$ is the composition of all $M_l(\cdot, \cdot, \cdot; \boldsymbol{\theta}_l^M)$ and $U_l(\cdot, \cdot; \boldsymbol{\theta}_l^U)$ at different layers. Since $\mathcal{N}^L(v)$ typically contains $\mathcal{O}(d^L)$ nodes, replaying $n$ nodes requires storing $\mathcal{O}(nd^L)$ nodes (the edges are not counted yet), where $d$ is the average degree. Therefore, the buffer size will be easily intractable in practice (*e.g.* the example of Reddit dataset in Introduction), and directly storing the computation ego-subgraphs for memory replay is infeasible for GNNs.

## 3.3 PARAMETER DECOUPLED GNNs WITH TEM

As we discussed earlier, the key challenge of applying memory replay to graph data is to preserve the rich topological information of the computation ego-subgraphs with potentially unbounded sizes. Therefore, a natural resolution is to preserve the crucial topological information with a compact vector such that the memory consumption is tractable. Formally, the desired subgraph representation can be defined as *Topology-aware Embedding* (TE).

**Definition 1** (Topology-aware embedding)**.** *Given a specific GNN parameterized with $\boldsymbol{\theta}$ and an input $\mathcal{G}_v^{sub}$, an embedding vector $\mathbf{e}_v$ is a topology-aware embedding for $\mathcal{G}_v^{sub}$ with respect to this GNN, if optimizing $\boldsymbol{\theta}$ with $\mathcal{G}_v^{sub}$ or $\mathbf{e}_v$ for this specific GNN are equivalent, i.e. $\mathbf{e}_v$ contains all necessary topological information of $\mathcal{G}_v^{sub}$ for training this GNN.*

However, TEs cannot be directly derived from the MPNNs due to their interleaved neighborhood aggregation and feature transformations. According to Section 3.2, whenever the trainable parameters get updated, recalculating the representation of a node $v$ requires all nodes and edges in $\mathcal{G}_v^{sub}$. To resolve this issue, we formulate the Parameter Decoupled Graph Neural Networks (PDGNNs) framework, which decouples the trainable parameters from the individual nodes/edges. PDGNNs may not be the only feasible framework to derive TEs, but is the first attempt and is empirically effective. Given $\mathcal{G}_v^{sub}$, the prediction of node $v$ with PDGNNs consists of two steps. First, the topological

information of $\mathcal{G}_v^{sub}$ is encoded into an embedding $\mathbf{e}_v$ via the function $\mathrm{f}_{topo}(\cdot)$ without trainable parameters (instantiations of $\mathrm{f}_{topo}(\cdot)$ are detailed in Section 3.4).

$$\mathbf{e}_v = \mathrm{f}_{topo}(\mathcal{G}_v^{sub}). \tag{4}$$

Next, $\mathbf{e}_v$ is further passed into a trainable function $\mathrm{f}_{out}(\cdot; \boldsymbol{\theta})$ parameterized by $\boldsymbol{\theta}$ (instantiations of $\mathrm{f}_{out}(\cdot; \boldsymbol{\theta})$ are detailed in Section 3.4) to get the output prediction $\hat{\mathbf{y}}_v$,

$$\hat{\mathbf{y}}_v = \mathrm{f}_{out}(\mathbf{e}_v; \boldsymbol{\theta}). \tag{5}$$

With the formulations above, $\mathbf{e}_v$ derived in Eq. (4) clearly satisfies the requirements of TE (Definition 1). Specifically, since the trainable parameters acts on $\mathbf{e}_v$ instead of any individual node/edge, optimizing the model parameters $\boldsymbol{\theta}$ with either $\mathbf{e}_v$ or $\mathcal{G}_v^{sub}$ are equivalent. Therefore, to retrain the model, the memory buffer only needs to store TEs instead of the original computation ego-subgraphs, which reduces the space complexity from $\mathcal{O}(nd^L)$ to $\mathcal{O}(n)$. We name the buffer to store the TEs as *Topology-aware Embedding Memory* ($\mathcal{TEM}$). Given a new task $\tau$, the update of $\mathcal{TEM}$ is:

$$\mathcal{TEM} = \mathcal{TEM} \bigcup \mathrm{sampler}(\{\mathbf{e}_v \mid v \in \mathbb{V}_\tau\}, n), \tag{6}$$

where $\mathrm{sampler}(\cdot, \cdot)$ is the adopted sampling strategy to populate the buffer, $\bigcup$ denotes the set union, and $n$ is the budget. According to the experimental results (Section 4.3), as long as $\mathcal{TEM}$ is maintained, PDGNNs-TEM can perform reasonably well with different choices of $\mathrm{sampler}(\cdot, \cdot)$, including the random sampling. Nevertheless, in Section 3.6, based on the theoretical insights in Section 3.5, we propose a novel sampling strategy to better populate $\mathcal{TEM}$ when the memory budget is tight, which is empirically verified to be effective in Section 4.3. Besides, Equation (6) assumes that all data of the current task are presented concurrently. In practice, the data of a task may come in multiple batches (*e.g.*, nodes come in batches on large graphs), and the buffer update have to be slightly modified, as detailed in Appendix C.1. For task $\tau$ with graph $\mathcal{G}_\tau$, the loss with $\mathcal{TEM}$ then becomes:

$$\mathcal{L} = \underbrace{\sum_{v \in \mathbb{V}_\tau} l(\mathrm{f}_{out}(\mathbf{e}_v; \boldsymbol{\theta}), \mathbf{y}_v)}_{\mathcal{L}_\tau: \text{ loss of the current task } \tau} + \lambda \underbrace{\sum_{\mathbf{e}_w \in \mathcal{TEM}} l(\mathrm{f}_{out}(\mathbf{e}_w; \boldsymbol{\theta}), \mathbf{y}_w)}_{\mathcal{L}_{aux}: \text{ auxiliary loss}}. \tag{7}$$

$\lambda$ balances the contribution of the data from the current task and the memory, and is typically manually chosen in traditional continual learning works. However, on graph data, we adopt a different strategy to re-scale the losses according to the class sizes to counter the bias from the severe class imbalance, which cannot be handled on graphs by directly balancing the datasets (details are provided in Appendix B.7).

## 3.4 INSTANTIATIONS OF PDGNNS

Although without trainable parameters, the function $\mathrm{f}_{topo}(\cdot)$ for generating TEs can be highly expressive with various formulations including linear and non-linear ones, both of which are studied in this work. First, the linear instantiations of $\mathrm{f}_{topo}(\cdot)$ can be generally formulated as,

$$\mathbf{e}_v = \mathrm{f}_{topo}(\mathcal{G}_v^{sub}) = \sum_{w \in \mathbb{V}} \mathbf{x}_w \cdot \pi(v, w; \hat{\mathbf{A}}), \tag{8}$$

where $\pi(\cdot, \cdot; \hat{\mathbf{A}})$ denotes the strategy for computation ego-subgraph construction and determines how would the model capture the topological information. Equation 8 describes the operation on each node. In practice, Equation 8 could be implemented as matrix multiplication to generate TEs of a set of nodes $\mathbb{V}$ in parallel, *i.e.* $\mathbf{E}_\mathbb{V} = \mathbf{\Pi}\mathbf{X}_\mathbb{V}$, where each entry $\mathbf{\Pi}_{v,w} = \pi(v, w; \hat{\mathbf{A}})$. $\mathbf{E}_\mathbb{V} \in \mathbb{R}^{|\mathbb{V}| \times b}$ is the concatenation of all TEs ($\mathbf{e}_v \in \mathbb{R}^b$), and $\mathbf{X}_\mathbb{V} \in \mathbb{R}^{|\mathbb{V}| \times b}$ is the concatenation of all node feature vectors $\mathbf{x}_v \in \mathbb{R}^b$. In our experiments, we adopt three representative strategies. The first strategy (S1) (Wu et al., 2019a) is a basic version of message passing and can be formulated as $\mathbf{\Pi} = \hat{\mathbf{A}}^L$. The second strategy (S2) (Zhu & Koniusz, 2020) considers balancing the contribution of neighborhood information from different hops via a hyperparameter $\alpha$, *i.e.* $\mathbf{\Pi} = \frac{1}{L} \sum_{l=1}^{L} \left( (1-\alpha)\hat{\mathbf{A}}^l + \alpha\mathbf{I} \right)$. Finally, we also adopt a strategy (S3) (Klicpera et al., 2018) that adjusts the contribution of the neighbors based on PageRank (Page et al., 1999), *i.e.* $\mathbf{\Pi} = \left( (1-\alpha)\hat{\mathbf{A}} + \alpha\mathbf{I} \right)^L$, in which $\alpha$ also balances the contribution of the neighborhood information.

The linear formulation of $\mathrm{f}_{topo}(\cdot)$ (Equation (8)) yields both promising experimental results (Section B) and instructive theoretical results (Section 3.5, and 3.6). Equation (8) is also highly efficient especially for large graphs due to the absence of iterative neighborhood aggregations. But $\mathrm{f}_{topo}(\cdot)$

can also take non-linear forms with more complex mappings, *e.g.*, the reservoir computing modules (Gallicchio & Micheli, 2020). The corresponding experimental and theoretical effects are introduced in Appendix B.2 and A.3.

Since $f_{out}(\cdot; \boldsymbol{\theta})$ simply deals with individual vectors (TEs), it is instantiated as MLP in this work. The specific configurations of $f_{out}(\cdot; \boldsymbol{\theta})$ is described in the experimental part (Section 4.2).

### 3.5 PSEUDO-TRAINING EFFECTS OF TES

In traditional continual learning on Euclidean data without explicit topological connections, replaying an example $\mathbf{x}_i$ (*e.g.*, an image) only reinforces the prediction of $\mathbf{x}_i$ itself. In this sub-section, we introduce the pseudo-training effect, which implies that training PDGNNs with $\mathbf{e}_v$ of node $v$ also influences the predictions of the other nodes in $\mathcal{G}_v^{sub}$, based on which we develop a novel sampling strategy to further boost the performance with a tight memory budget.

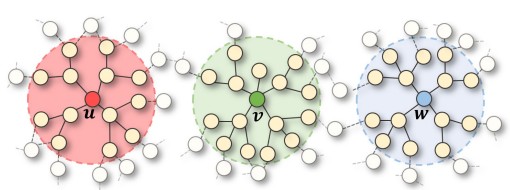

Figure 2: Illustration of the coverage ratio. Supposing the graph has $N$ nodes, $R_c(\{u\}) = \frac{13}{N}$, $R_c(\{v\}) = \frac{15}{N}$, $R_c(\{u\}) = \frac{14}{N}$, and $R_c(\{u, v, w\}) = \frac{42}{N}$

**Theorem 1** (Pseudo-training). *Given a node $v$, its computation ego-subgraph $\mathcal{G}_v^{sub}$, the TE $\mathbf{e}_v$, and label $\mathbf{y}_v$ (suppose $v$ belongs to class $k$, i.e. $\mathbf{y}_{v,k} = 1$), then training PDGNNs with $\mathbf{e}_v$ has the following two properties:*

*1. It is equivalent to training PDGNNs with each node $w$ in $\mathcal{G}_v^{sub}$ with $\mathcal{G}_v^{sub}$ being a pseudo computation ego-subgraph and $\mathbf{y}_v$ being a pseudo label, where the contribution of $\mathbf{x}_w$ (via Equation 8) is re-scaled by $\frac{\pi(v,w;\hat{\mathbf{A}})}{\pi(w,w;\hat{\mathbf{A}})}$. We term this property as the pseudo-training effect on neighboring nodes, because it is equivalent to that the training is conducted on each neighboring node (in $\mathcal{G}_v^{sub}$) through the pseudo labels and the pseudo computation ego-subgraphs.*

*2. When $f_{out}(\cdot; \boldsymbol{\theta})$ is linear, training PDGNNs on $\mathbf{e}_v$ is also equivalent to training $f_{out}(\cdot; \boldsymbol{\theta})$ on pseudo-labeled nodes $(\mathbf{x}_w, \mathbf{y}_v)$ for each $w$ in $\mathcal{G}_v^{sub}$, where the contribution of $w$ in the loss is adaptively re-scaled with a weight $\frac{f_{out}(\mathbf{x}_w; \boldsymbol{\theta})_k \cdot \pi(v,w;\hat{\mathbf{A}})}{\sum_{w \in \mathbb{V}_v^{sub}} f_{out}\left(\mathbf{x}_w \cdot \pi(v,w;\hat{\mathbf{A}}); \boldsymbol{\theta}\right)_k}$.*

The pseudo-training effect essentially arises from the neighborhood aggregation operation of GNNs, of which the rationale is to iteratively refine the node embeddings with similar neighbors. Pseudo-training effect implies that replaying the TE of one node can also strengthen the prediction for its neighbors within the same computation ego-subgraph and alleviate the forgetting problem on them. More detailed discusion on pseudo-training effect is provided in Appendix A.3 and Appendix A.4. The above analysis suggests that TEs with larger computation ego-subgraphs covering more nodes may be more effective, motivating our coverage maximization sampling strategy in the next subsection, which is also empirically justified in Section 4.3.

### 3.6 COVERAGE MAXIMIZATION SAMPLING

Following the above subsection, TEs with larger computation ego-subgraphs are preferred to be stored. To quantify the size of the computation ego-subgraphs, we formally define the coverage ratio of the selected TEs as the nodes covered by their computation ego-subgraphs versus the total nodes in the graph (Figure 2). Since a TE uniquely corresponds to a node, we may use 'node' and 'TE' interchangeably.

**Definition 2.** *Given a graph $\mathcal{G}$, node set $\mathbb{V}$, and function $\pi(\cdot, \cdot; \hat{\mathbf{A}})$, the coverage ratio of a set of nodes $\mathbb{V}_s$ is:*

$$R_c(\mathbb{V}_s) = \frac{|\cup_{v \in \mathbb{V}_s} \{w | w \in \mathcal{G}_v^{sub}\}|}{|\mathbb{V}|}, \qquad (9)$$

*i.e., the ratio of nodes of the entire (training) graph covered by the computation ego-subgraphs of the selected nodes (TEs).*

---

**Algorithm 1** Coverage maximization sampling

**Input:** $\mathcal{G}_\tau$, $\mathbb{V}_\tau$, $\hat{\mathbf{A}}_\tau$, $\pi(\cdot, \cdot; \cdot)$, sample size $n$.
**Output:** Selected nodes $\mathcal{S}$
1: Initialize $\mathcal{S} = \{\}$.
2: **for each** $v \in \mathbb{V}_\tau$ **do**
3: $\quad R_c(\{v\}) = \frac{|\{w | w \in \mathcal{G}_{\tau,v}^{sub}\}|}{|\mathbb{V}_\tau|}$
4: **end for each**
5: **for each** $v \in \mathbb{V}_\tau$ **do**
6: $\quad p_v = \frac{R_c(\{v\})}{\sum_{w \in \mathbb{V}_\tau} R_c(\{w\})}$
7: **end for each**
8: **while** $n > 0$ **do**
9: $\quad$ Sample one node $v$ from $\mathbb{V}_\tau$ according to $\{p_w \mid w \in \mathbb{V}_\tau\}$.
10: $\quad \mathcal{S} = \mathcal{S} \cup \{v\}$
11: $\quad \mathbb{V}_\tau = \mathbb{V}_\tau \backslash \{v\}$ $\qquad \triangleright$ Sampling without replacement
12: $\quad n \leftarrow n - 1$
13: **end while**

---

Table 1: Performance & coverage ratios of different sampling strategies and buffer sizes on OGB-Arxiv dataset ($\uparrow$ higher means better).

| Ratio of dataset /% | | 0.02 | 0.1 | 1.0 | 5.0 | 40.0 |
|---|---|---|---|---|---|---|
| AA/% | Uniform samp. | 12.0±1.1 | 24.1±1.7 | 42.2±0.3 | 50.4±0.4 | 53.3±0.4 |
| | Mean of feat. | 12.6±0.1 | 25.3±0.3 | 42.8±0.3 | 50.4±0.7 | 53.3±0.2 |
| | Cov. Max. | **14.9±0.8** | **26.8±1.8** | **43.7±0.5** | **50.5±0.4** | **53.4±0.1** |
| Cov. ratio/% | Uniform samp. | 0.1±0.1 | 0.3±0.0 | 3.5±0.9 | 15.9±1.1 | 84.8±1.5 |
| | Mean of feat. | 0.2±0.4 | 0.6±0.3 | 7.1±0.6 | 29.6±1.7 | 91.1±0.1 |
| | Cov. Max. | **0.5±1.1** | **2.9±1.8** | **22.5±1.6** | **46.3±0.6** | **92.8±0.0** |

Table 2: Performance comparisons under class-IL on different datasets ($\uparrow$ higher means better).

| C.L.T. | CoraFull | | OGB-Arxiv | | Reddit | | OGB-Products | |
|---|---|---|---|---|---|---|---|---|
| | AA/% $\uparrow$ | AF/% $\uparrow$ | AA/% $\uparrow$ | AF /% $\uparrow$ | AA/% $\uparrow$ | AF /% $\uparrow$ | AA/% $\uparrow$ | AF /% $\uparrow$ |
| Fine-tune | 3.5±0.2 | -95.2±0.3 | 4.9±0.0 | -89.7±0.4 | 5.9±0.9 | -97.9±1.8 | 7.6±0.6 | -88.7±0.5 |
| EWC 2017 | 52.6±5.0 | -38.5±7.3 | 8.5±0.6 | -69.5±4.7 | 10.3±6.3 | -33.2±14.6 | 23.8±2.2 | -21.7±3.9 |
| MAS 2018 | 6.5±1.0 | -92.3±1.0 | 4.8±0.2 | -72.2±2.6 | 9.2±7.6 | -23.1±14.6 | 16.7±2.6 | -57.0±24.8 |
| GEM 2017 | 7.4±0.1 | -91.0±0.1 | 4.9±0.0 | -89.8±0.2 | 5.0±0.0 | -99.4±0.0 | 4.5±0.8 | -94.7±0.2 |
| TWP 2021 | 62.6±1.7 | -30.6±2.8 | 6.7±1.2 | -50.6±6.9 | 8.0±2.9 | -18.8±5.1 | 14.1±2.1 | -11.4±1.3 |
| LwF 2017 | 33.4±0.9 | -59.6±1.2 | 9.9±6.7 | -43.6±7.5 | 86.6±0.8 | -9.2±0.9 | 48.2±0.8 | -18.6±0.9 |
| ER-GNN 2021 | 2.9±0.0 | -94.6±0.1 | 12.3±3.1 | -79.9±3.3 | 20.4±2.6 | -82.7±2.9 | 56.7±0.3 | -33.3±0.5 |
| SSM 2022d | 75.4±0.1 | -9.7±0.0 | 48.3±0.5 | -10.7±0.3 | 94.4±0.0 | -1.3±0.0 | 63.3±0.1 | -9.6±0.3 |
| Joint | 80.8±0.1 | -3.1±0.2 | 56.8±0.0 | -8.6±0.0 | 97.1±0.1 | -0.7±0.1 | 71.5±0.1 | -5.8±0.2 |
| PDGNNs | **81.9±0.1** | **-3.9±0.1** | **53.2±0.2** | **-14.7±0.2** | **96.6±0.0** | **-2.6±0.1** | **73.9±0.1** | **-10.9±0.2** |

To maximize $R_c(\mathcal{TEM})$, a naive approach is to start from selecting the TE with the largest coverage ratio, and then iteratively incorporate TE that increases $R_c(\mathcal{TEM})$ the most. However, this requires computing $R_c(\mathcal{TEM})$ for all candidate TEs at each iteration, which is time consuming especially on large graphs. Besides, certain randomness is also desired for the diversity of $\mathcal{TEM}$. Therefore, we propose to sample TEs based on their coverage ratio. Specifically, in task $\tau$, the probability of sampling node $v \in \mathbb{V}_\tau$ is $p_v = \frac{R_c(\{v\})}{\sum_{w \in \mathbb{V}_\tau} R_c(\{w\})}$. Then the nodes in $\mathbb{V}_\tau$ are sampled according to $\{p_v \mid v \in \mathbb{V}_\tau\}$ without replacement, as shown in Algorithm 1. In experiments, we demonstrate the correlation between the coverage ratio and the performance, which verifies the benefits revealed in Section 3.5

## 4 EXPERIMENTS

In this section, we aim to answer the following research questions: RQ1: Whether PDGNNs-TEM works well with a reasonable buffer size? RQ2: Does coverage maximization sampling ensure a higher coverage ratio and better performance when the memory budget is tight? RQ3: Whether our theoretical results can be reflected in experiments? RQ4: Whether PDGNNs-TEM can outperform the state-of-the-art methods in both class-IL and task-IL scenarios? Due to the space limitations, only the most prominent results are presented in the main content, and more details are available in Appendix B. For simplicity, PDGNNs-TEM will be denoted as PDGNNs in this section.

### 4.1 DATASETS

Following the public benchmark CGLB (Zhang et al., 2022b), we adopted four datasets, CoraFull, OGB-Arxiv, Reddit, and OGB-Products, with up to millions of nodes and 70 classes. Dataset statistics and task splittings are summarized in Table 8. In the paper, we show the results under the splittings with the largest number of tasks. More details of the datasets and results with additional configurations are provided in the Appendix B.6 and B.3, respectively.

### 4.2 EXPERIMENTAL SETUP AND MODEL EVALUATION

**Continual learning setting and model evaluation.** During training, a model is trained on a task sequence. During testing, the model is tested on all learned tasks. Class-IL scenario requires a model to classify a given node by picking a class from all learned classes (more challenging), while task-IL scenario only requires the model to distinguish the classes within each task. For model evaluation, the most thorough metric is the accuracy matrix $M^{acc} \in \mathbb{R}^{T \times T}$, where $M_{i,j}^{acc}$ denotes the accuracy on task $j$ after learning task $i$. The learning dynamics can be reflected with average accuracy (AA) over all learnt tasks after learning each new task, *i.e.*, $\left\{ \frac{\sum_{j=1}^{i} M_{i,j}^{acc}}{i} \mid i = 1, ..., T \right\}$, which can be visualized as a curve. Similarly, the average forgetting (AF) after learning each task can reflect the learning dynamics

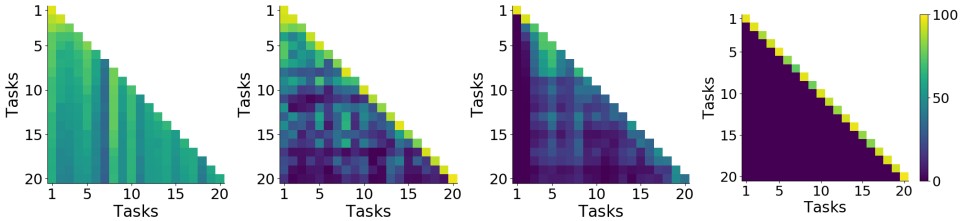

Figure 3: Dynamics of average accuracy in the class-IL scenario.(a) CoraFull, 2 classes per task. (b) OGB-Arxiv, 2 classes per task. (c) Reddit, 2 classes per task. (d) OGB-Products, 2 classes per task.

Figure 4: From left to right: accuracy matrix of PDGNNs, ER-GNN, LwF, and Fine-tune on OGB-Arxiv dataset.

from the perspective of forgetting, $\left\{\frac{\sum_{j=1}^{i-1} \mathrm{M}_{i,j}^{acc} - \mathrm{M}_{j,j}^{acc}}{i-1} | i = 2, ..., T\right\}$. To use a single numeric value for evaluation, the AA and AF after learning all $T$ tasks will be used. These two metrics are widely adopted in continual learning works (Chaudhry et al., 2018; Lopez-Paz & Ranzato, 2017; Liu et al., 2021; Zhang et al., 2022c; Zhou & Cao, 2021), although the names are different in different works. We repeat all experiments 5 times on one Nvidia Titan Xp GPU. All results are reported with average performance and standard deviations.

**Baselines and model settings.** Our baselines for continual learning on expanding graphs include Experience Replay based GNN (ER-GNN) (Zhou & Cao, 2021), Topology-aware Weight Preserving (TWP) (Liu et al., 2021) and Sparsified Subgraph Memory (SSM) (Zhang et al., 2022d). Milestone works for Euclidean data but also applicable to GNNs include Elastic Weight Consolidation (EWC) (Kirkpatrick et al., 2017), Learning without Forgetting (LwF) (Li & Hoiem, 2017), Gradient Episodic Memory (GEM) (Lopez-Paz & Ranzato, 2017), and Memory Aware Synapses (MAS) (Aljundi et al., 2018)), are also adopted. HPNs (Zhang et al., 2022c) is designed to work under a stricter task-IL setting, and cannot be properly incorporated for comparison. These baselines are implemented based on three popular backbone GNNs, *i.e.*, Graph Convolutional Network (GCN) (Kipf & Welling, 2017), Graph Attentional Network (GAT) (Veličković et al., 2017), and Graph Isomorphism Network (GIN) (Xu et al., 2018). Besides, joint training (without forgetting problem) and fine-tune (without continual learning technique) are adopted as the upper and lower bound on the performance. We instantiate $\mathrm{f}_{out}(\cdot; \boldsymbol{\theta})$ as a multi-layer perceptron (MLP). For a fair comparison, all methods including $\mathrm{f}_{out}(\cdot; \boldsymbol{\theta})$ of PDGNNs are set as 2-layer with 256 hidden dimensions, and $L$ in Section 3.3 is set as 2 for consistency. Additional discussion on the model depth is provided in Appendix B.5. As detailed in Section 4.3, $\mathrm{f}_{topo}(\cdot)$ is chosen as strategy S1 (Section 3.4), while the comparison among different choices are introduced in Appendix B.2.

### 4.3 STUDIES ON THE BUFFER SIZE & PERFORMANCE VS. COVERAGE RATIO (RQ1, 2, AND 3)

In Table 1, based on PDGNNs, we compare the proposed *coverage maximization sampling* with uniform sampling and mean of feature (MoF) sampling in terms of coverage ratios and performance when the buffer size (ratio of the dataset) varies from 0.0002 to 0.4 on the OGB-Arxiv dataset. Our proposed *coverage maximization sampling* achieves a superior coverage ratio, which indeed enhances the performance when the memory budget is tight. In real-world applications, a tight memory budget is a very common situation, making the coverage maximization sampling a favorable choice. We also notice that the average accuracy for *coverage maximization sampling* is positively related to the coverage ratio in general, which is consistent with the Theorem 2. Table 1 also demonstrates the high memory efficiency of TEM. No matter which sampling strategy is used, the performance can reach ≈50% average accuracy (AA) with only 5% data buffered. In Appendix B.2, we further evaluate how the performance changes when the buffer size varies with different variants of PDGNNs. (*i.e.*, the TE generation strategies adopted from SGC, $S^2$GC, APPNP, and reservoir computing described in Section 3.3). In Appendix B.4, we provide the comparison of the space consumption of different memory based strategies to demonstrate the efficiency of PDGNNs-TEM. In Appendix C.2, we added

Table 3: Performance comparisons under task-IL on different datasets ($\uparrow$ higher means better).

| C.L.T. | CoraFull | | OGB-Arxiv | | Reddit | | OGB-Products | |
|---|---|---|---|---|---|---|---|---|
| | AA/% $\uparrow$ | AF/% $\uparrow$ | AA/% $\uparrow$ | AF /% $\uparrow$ | AA/% $\uparrow$ | AF /% $\uparrow$ | AA/% $\uparrow$ | AF /% $\uparrow$ |
| Fine-tune | 56.0±2.4 | -41.0±2.5 | 56.2±1.7 | -36.2±1.7 | 79.5±13.0 | -11.7±2.9 | 64.4±2.3 | -31.1±2.6 |
| EWC 2017 | 89.8±0.7 | -5.1±0.3 | 71.5±0.4 | -0.9±0.4 | 83.9±11.9 | -2.0±1.0 | 87.0±0.9 | -1.7±0.9 |
| MAS 2018 | 92.2±0.6 | -3.7±0.8 | 72.7±1.6 | -18.5±1.6 | 61.1±4.3 | -0.5±0.7 | 80.6±2.4 | -13.7±2.4 |
| GEM 2017 | 92.0±0.4 | 0.3±0.8 | 80.8±0.8 | -5.3±0.9 | **98.9±0.0** | **-0.5±0.1** | 87.7±1.1 | -7.0±1.2 |
| TWP 2021 | 94.3±0.5 | -1.6±0.3 | 80.9±1.0 | -1.3±0.8 | 78.0±13.4 | -0.2±0.3 | 81.8±2.2 | -0.3±0.5 |
| LwF 2017 | 93.8±0.1 | -0.4±0.1 | 71.1±1.7 | -1.5±0.5 | 98.6±0.1 | -0.0±0.0 | 86.3±0.1 | -0.5±0.1 |
| ER-GNN 2021 | 62.4±1.5 | -34.5±1.5 | 86.4±0.2 | 0.5±0.3 | 97.5±1.5 | 2.6±3.7 | 86.4±0.0 | 11.7±0.0 |
| SSM 2022d | 94.1±0.4 | 0.5±0.3 | 87.1±0.5 | -1.4±0.2 | 98.8±0.1 | -0.2±0.3 | 88.8±0.6 | -1.3±0.3 |
| Joint | 96.0±0.1 | 0.0±0.1 | 90.3±0.2 | 0.5±0.2 | 99.5±0.0 | 0.0±0.0 | 95.3±0.4 | -0.3±0.3 |
| **PDGNNs** | **94.6±0.1** | **0.6±1.0** | **89.8±0.4** | **-0.0±0.5** | **98.9±0.0** | **-0.5±0.0** | **93.5±0.5** | **-2.1±0.1** |

discussions to differentiate the coverage maximization with several essentially different works with similar names (Chen et al., 2009; Wang et al., 2016; Zhou & Cao, 2021).

### 4.4 RESULTS FOR CLASS-IL SCENARIO AND TASK-IL SCENARIO (RQ4)

**Class-IL Scenario**. As shown in Table 2, under the class-IL scenario, PDGNNs significantly outperform the baselines and are even comparable to joint training (the performance upper bound) on all 4 public datasets. The learning dynamics are shown in Figure 3. Since the curve of PDGNNs is very close to the curve of joint training, we conclude that the forgetting problem is nearly eliminated by PDGNNs. In Table 2 and Figure 3, PDGNNs sometimes outperform joint training. The reasons are two-fold. First, PDGNNs learn the tasks sequentially while joint training optimizes the model for all tasks simultaneously, resulting in different optimization difficulties (Bhat et al., 2021). Second, when learning new tasks, joint training accesses all previous data that may be noisy, while replaying the representative TEs may help filter out noise. More discussions are provided in Appendix B.3.

To thoroughly understand different methods, we visualize the accuracy matrices of four representative methods, including our PDGNNs (memory replay with topological information), ER-GNN (memory replay without topological information), LwF (relatively satisfying performance without memory buffer), and Fine-tune (without continual learning technique), in Figure 4. Each row of the matrix denotes the performance on each learnt task after learning a new task, and each column denotes the performance change of a specific task. Compared to the baselines, PDGNNs maintain stable performance on each task even though new tasks are continuously learned. Besides, we also visualized the learnt node representations at different learning stages in Appendix B.1.

**Task-IL Scenario.** The comparison results under the task-IL scenario are shown in Table 3. We can observe that PDGNNs still consistently outperform baselines on all different datasets, even though task-IL is less challenging than the class-IL as we discussed in Section 4.2.

### 4.5 MEMORY CONSUMPTION OF DIFFERENT METHODS

In this subsection, we compare the memory space consumption of different memory based methods to concretely demonstrate the memory efficiency of PDGNNs-TEM. The final memory consumption (measured by the number of float32 values) after learning each entire dataset are shown in Table 4. As a reference, the memory consumption of storing full computation ego-subgraph is also calculated.

| C.L.T. | CoraFull | OGB-Arxiv | Reddit | OGB-Products |
|---|---|---|---|---|
| Full Subgraph | 7,264M | 35M | 2,184,957M | 5,341M |
| GEM 2017 | 7,840M | 86M | 329M | 82M |
| ER-GNN 2021 | 61M | 2M | 12M | 3M |
| SSM 2022d | 732M | 41M | 193M | 37M |
| PDGNNs-TEM | 37M | 2M | 9M | 2M |

Table 4: Memory space consumption of different methods.

### 5 CONCLUSION

In this work, we propose the PDGNNs with TEM framework for continual learning on expanding graphs. Based on TEs, we reduce the memory space complexity from $\mathcal{O}(nd^L)$ to $\mathcal{O}(n)$, which enables PDGNNs to fully utilize the explicit topological information sampled from the previous tasks. We also discover and theoretically analyze the pseudo-training effect of TEs. This inspires us to develop the *coverage maximization sampling* which has been demonstrated to be highly efficient when the memory budget is tight. Finally, thorough empirical studies on both class-IL and task-IL continual learning scenarios demonstrate the effectiveness of PDGNNs-TEM.

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

# A    THEORETICAL ANALYSIS

In this section, we give proofs and detailed analysis of the theoretical results in the paper.

## A.1    PARAMETER DECOUPLED GNNS WITH TEM

In Section 3.3 of the paper, we mentioned that the embedding $\mathbf{e}_v$ derived in PDGNNs is a topology-aware embedding of $\mathcal{G}_v^{sub}$ with respect to the optimization of $\boldsymbol{\theta}$. Although this is intuitive, we still provide a formal proof for it.

*Proof.* According to Definition 1 in the paper, a sufficient condition for a vector $\mathbf{e}_v$ to be a topology-aware embedding of $\mathcal{G}_v^{sub}$ is that $\mathbf{e}_v$ provides same information as $\mathcal{G}_v^{sub}$ for optimizing the parameter $\boldsymbol{\theta}$ of a model $\mathrm{f}_{out}(\cdot;\boldsymbol{\theta})$. Therefore, the proof can be done by showing $\nabla_{\boldsymbol{\theta}}\mathcal{L}(\mathbf{e}_v,\boldsymbol{\theta}) = \nabla_{\boldsymbol{\theta}}\mathcal{L}(\mathcal{G}_v^{sub},\boldsymbol{\theta})$, where $\mathcal{L}$ is the adopted loss function. This becomes straightforward under the PDGNNs framework since $\mathcal{G}_v^{sub}$ is first embedded in $\mathbf{e}_v$ and then participate in the computation with the trainable parameter $\boldsymbol{\theta}$. Specifically, given an input computation ego-subgraph $\mathcal{G}_v^{sub}$ with the label $\mathbf{y}_v$, the corresponding prediction of PDGNNs is:

$$\hat{y}_v = \mathrm{f}_{out}\Big( \sum_{w \in \mathbb{V}_v^{sub}} \mathbf{x}_w \cdot \pi(v, w; \hat{\mathbf{A}}); \boldsymbol{\theta}\Big), \tag{10}$$

and the loss is:

$$\mathcal{L}_v = l\Bigg( \mathrm{f}_{out}\Big( \sum_{w \in \mathbb{V}_v^{sub}} \mathbf{x}_w \cdot \pi(v, w; \hat{\mathbf{A}}); \boldsymbol{\theta}\Big), \mathbf{y}_v \Bigg), \tag{11}$$

the gradient of loss $\mathcal{L}_v$ is:

$$\nabla_{\boldsymbol{\theta}}\mathcal{L}_v = \nabla_{\boldsymbol{\theta}} l\Bigg( \mathrm{f}_{out}\Big( \sum_{w \in \mathbb{V}_v^{sub}} \mathbf{x}_w \cdot \pi(v, w; \hat{\mathbf{A}}); \boldsymbol{\theta}\Big), \mathbf{y}_v \Bigg). \tag{12}$$

When the input $\mathcal{G}_v^{sub}$ is replaced with $\mathbf{e}_v$, the prediction becomes:

$$\hat{\mathbf{y}}_v = \mathrm{f}_{out}(\mathbf{e}_v; \boldsymbol{\theta}), \tag{13}$$

and the corresponding loss becomes:

$$\mathcal{L}_v' = l\big(\mathrm{f}_{out}(\mathbf{e}_v; \boldsymbol{\theta}), \mathbf{y}_v\big), \tag{14}$$

the gradient of loss $\mathcal{L}_v$ becomes:

$$\nabla_{\boldsymbol{\theta}}\mathcal{L}_v' = \nabla_{\boldsymbol{\theta}} l\big(\mathrm{f}_{out}(\mathbf{e}_v; \boldsymbol{\theta}), \mathbf{y}_v\big). \tag{15}$$

Since in the PDGNNs, $\mathbf{e}_v$ is calculated as:

$$\mathbf{e}_v = \sum_{w \in \mathbb{V}_v^{sub}} \mathbf{x}_w \cdot \pi(v, w; \hat{\mathbf{A}}), \tag{16}$$

then we have:

$$\nabla_{\boldsymbol{\theta}}\mathcal{L}_v = \nabla_{\boldsymbol{\theta}}\mathcal{L}_v', \tag{17}$$

*i.e.*, optimizing the trainable parameters with $\mathbf{e}_v$ is equal to optimizing the trainable parameters with $\mathcal{G}_v^{sub}$. $\qquad\square$

## A.2    PSEUDO-TRAINING EFFECTS OF TES

**Theorem 2** (Pseudo-training). *Given a node $v$, its computation ego-subgraph $\mathcal{G}_v^{sub}$, the TE $\mathbf{e}_v$, and label $\mathbf{y}_v$ (suppose $v$ belongs to class $k$, i.e. $\mathbf{y}_{v,k} = 1$), then training PDGNNs with $\mathbf{e}_v$ has the following two properties:*

1. *It is equivalent to training PDGNNs with each node $w$ in $\mathcal{G}_v^{sub}$ with $\mathcal{G}_v^{sub}$ being a pseudo computation ego-subgraph and $\mathbf{y}_v$ being a pseudo label, where the contribution of $\mathbf{x}_w$ (via Equation*

*4 in the paper) is re-scaled by $\frac{\pi(v,w;\hat{\mathbf{A}})}{\pi(w,w;\hat{\mathbf{A}})}$. We term this property as the pseudo-training effect on neighboring nodes.*

2. *When* $f_{out}(\cdot;\boldsymbol{\theta})$ *is linear, training PDGNNs on* $\mathbf{e}_v$ *is also equivalent to training* $f_{out}(\cdot;\boldsymbol{\theta})$ *on pseudo-labeled nodes* $(\mathbf{x}_w, \mathbf{y}_v)$ *for each* $w$ *in* $\mathcal{G}_v^{sub}$, *where the contribution of* $w$ *in the loss is adaptively re-scaled with a weight* $\frac{f_{out}(\mathbf{x}_w;\boldsymbol{\theta})_k \cdot \pi(v,w;\hat{\mathbf{A}})}{\sum_{w\in\mathbb{V}_v^{sub}} f_{out}\left(\mathbf{x}_w\cdot\pi(v,w;\hat{\mathbf{A}});\boldsymbol{\theta}\right)_k}$.

*Proof of Theorem 2.1.* Given a node $v$, the prediction is:

$$\hat{\mathbf{y}}_v = f_{out}(\mathbf{e}_v; \boldsymbol{\theta}) \tag{18}$$

$\because \mathbf{e}_v = \sum_{w\in\mathbb{V}_v^{sub}} \mathbf{x}_w \cdot \pi(v, w; \hat{\mathbf{A}})$, where $\mathbb{V}_v^{sub}$ denotes the node set of the computation ego-subgraph $\mathcal{G}_v^{sub}$, and $\hat{\mathbf{A}}$ is the adjacency matrix of $\mathcal{G}_v^{sub}$.

$\therefore$

$$\hat{\mathbf{y}}_v = f_{out}\Big( \sum_{w\in\mathbb{V}_v^{sub}} \mathbf{x}_w \cdot \pi(v, w; \hat{\mathbf{A}}); \boldsymbol{\theta} \Big) \tag{19}$$

Given the target (ground truth label) of node $v$ as $y_v$, the objective function of training the model with node $v$ is formulated as:

$$\mathcal{L}_v = l\left( f_{out}\Big( \sum_{w\in\mathbb{V}_v^{sub}} \mathbf{x}_w \cdot \pi(v, w; \hat{\mathbf{A}}); \boldsymbol{\theta} \Big), \mathbf{y}_v \right), \tag{20}$$

where $l$ could be any loss function to measure the distance between the prediction and the target.

Since $\mathbb{V}_v^{sub}$ contains both the features of node $v$ and its neighbors, Equation 20 can be further expanded to separate the contribution of node $v$ and its neighbors:

$$\mathcal{L}_v = l\left( f_{out}\Big( \underbrace{\mathbf{x}_v \cdot \pi(v, v; \hat{\mathbf{A}})}_{\text{information from node } v} + \underbrace{\sum_{w\in\mathbb{V}_v^{sub}\setminus\{v\}} \mathbf{x}_w \cdot \pi(v, w; \hat{\mathbf{A}})}_{\text{neighborhood information}}; \boldsymbol{\theta} \Big), \mathbf{y}_v \right), \tag{21}$$

Given an arbitrary node $q \in \mathbb{V}_v^{sub}$ but $q \neq v \in \mathbb{V}_v^{sub}$ (the adjacency matrix $\hat{\mathbf{A}}$ stays the same), we can similarly obtain the loss of training the model with node $q$:

$$\mathcal{L}_q = l\left( f_{out}\Big( \underbrace{\mathbf{x}_q \cdot \pi(q, q; \hat{\mathbf{A}})}_{\text{information from node } q} + \underbrace{\sum_{w\in\mathbb{V}_q^{sub}\setminus\{q\}} \mathbf{x}_w \cdot \pi(q, w; \hat{\mathbf{A}})}_{\text{neighborhood information}}; \boldsymbol{\theta} \Big), \mathbf{y}_q \right). \tag{22}$$

Since $q \in \mathbb{V}_v^{sub}\setminus\{v\}$, we rewrite Equation 21 as:

$$\mathcal{L}_v = l\left( f_{out}\Big( \underbrace{\mathbf{x}_q \cdot \pi(v, q; \hat{\mathbf{A}})}_{\text{information from node } q} + \underbrace{\sum_{w\in\mathbb{V}_v^{sub}\setminus\{q\}} \mathbf{x}_w \cdot \pi(v, w; \hat{\mathbf{A}})}_{\text{neighborhood information}}; \boldsymbol{\theta} \Big), \mathbf{y}_v \right), \tag{23}$$

By comparing Equation 23 and 22, we could observe the similarity in the loss of node $v$ and $q$, and the difference lies in the contribution (weight $\pi(\cdot, \cdot; \hat{\mathbf{A}})$) of each node and the neighboring nodes ($\mathbb{V}_q^{sub}$ and $\mathbb{V}_v^{sub}$). $\qquad\square$

*Proof of Theorem 2.2.* In this part, we choose the loss function $l$ as cross entropy $CE(\cdot, \cdot)$, which is the common choice for classification problems. In the following, we will first derive the gradient of training the PDGNNs with $(\mathbf{e}_v, y_v)$. For cross entropy, we denote the one-hot vector form label as $\mathbf{y}_v$,

of which the $y_v$-th element is one and other entries are zero. Given the loss of a node $v$ as shown in the Equation 20, the gradient is derived as:

$$\nabla_{\boldsymbol{\theta}} \mathcal{L}_v = \nabla_{\boldsymbol{\theta}} \mathrm{CE}\left( \sum_{w \in \mathbb{V}_v^{sub}} \mathrm{f}_{out}\left(\mathbf{x}_w \cdot \pi(v, w; \hat{\mathbf{A}}); \boldsymbol{\theta}\right), \mathbf{y}_v \right) \tag{24}$$

$$= \nabla_{\boldsymbol{\theta}} \left( \mathbf{y}_{v,k} \cdot \log \sum_{w \in \mathbb{V}_v^{sub}} \mathrm{f}_{out}\left(\mathbf{x}_w \cdot \pi(v, w; \hat{\mathbf{A}}); \boldsymbol{\theta}\right)_k \right) \tag{25}$$

$$= \mathbf{y}_{v,k} \cdot \frac{\nabla_{\boldsymbol{\theta}} \left( \sum_{w \in \mathbb{V}_v^{sub}} \mathrm{f}_{out}\left(\mathbf{x}_w \cdot \pi(v, w; \hat{\mathbf{A}}); \boldsymbol{\theta}\right)_k \right)}{\sum_{w \in \mathbb{V}_v^{sub}} \mathrm{f}_{out}\left(\mathbf{x}_w \cdot \pi(v, w; \hat{\mathbf{A}}); \boldsymbol{\theta}\right)_k} \tag{26}$$

$$= \mathbf{y}_{v,k} \cdot \frac{\sum_{w \in \mathbb{V}_v^{sub}} \nabla_{\boldsymbol{\theta}} \mathrm{f}_{out}\left(\mathbf{x}_w \cdot \pi(v, w; \hat{\mathbf{A}}); \boldsymbol{\theta}\right)_k}{\sum_{w \in \mathbb{V}_v^{sub}} \mathrm{f}_{out}\left(\mathbf{x}_w \cdot \pi(v, w; \hat{\mathbf{A}}); \boldsymbol{\theta}\right)_k} \tag{27}$$

$$= \mathbf{y}_{v,k} \cdot \frac{\sum_{w \in \mathbb{V}_v^{sub}} \nabla_{\boldsymbol{\theta}} \mathrm{f}_{out}(\mathbf{x}_w; \boldsymbol{\theta})_k \cdot \pi(v, w; \hat{\mathbf{A}})}{\sum_{w \in \mathbb{V}_v^{sub}} \mathrm{f}_{out}\left(\mathbf{x}_w \cdot \pi(v, w; \hat{\mathbf{A}}); \boldsymbol{\theta}\right)_k} \tag{28}$$

$$= \frac{\sum_{w \in \mathbb{V}_v^{sub}} \mathbf{y}_{v,k} \cdot \frac{\nabla_{\boldsymbol{\theta}} \mathrm{f}_{out}(\mathbf{x}_w; \boldsymbol{\theta})_k}{\mathrm{f}_{out}(\mathbf{x}_w; \boldsymbol{\theta})_k} \cdot \mathrm{f}_{out}(\mathbf{x}_w; \boldsymbol{\theta})_k \cdot \pi(v, w; \hat{\mathbf{A}})}{\sum_{w \in \mathbb{V}_v^{sub}} \mathrm{f}_{out}\left(\mathbf{x}_w \cdot \pi(v, w; \hat{\mathbf{A}}); \boldsymbol{\theta}\right)_k} \tag{29}$$

$$= \frac{\sum_{w \in \mathbb{V}_v^{sub}} \nabla_{\boldsymbol{\theta}} \mathrm{CE}\left(\mathrm{f}_{out}(\mathbf{x}_w; \boldsymbol{\theta}), \mathbf{y}_{v,k}\right) \cdot \mathrm{f}_{out}(\mathbf{x}_w; \boldsymbol{\theta}) \cdot \pi(v, w; \hat{\mathbf{A}})}{\sum_{w \in \mathbb{V}_v^{sub}} \mathrm{f}_{out}\left(\mathbf{x}_w \cdot \pi(v, w; \hat{\mathbf{A}}); \boldsymbol{\theta}\right)} \tag{30}$$

$$= \sum_{w \in \mathbb{V}_v^{sub}} \frac{\mathrm{f}_{out}(\mathbf{x}_w; \boldsymbol{\theta}) \cdot \pi(v, w; \hat{\mathbf{A}})}{\sum_{w \in \mathbb{V}_v^{sub}} \mathrm{f}_{out}\left(\mathbf{x}_w \cdot \pi(v, w; \hat{\mathbf{A}}); \boldsymbol{\theta}\right)} \cdot \nabla_{\boldsymbol{\theta}} \mathrm{CE}\left(\mathrm{f}_{out}(\mathbf{x}_w; \boldsymbol{\theta}), \mathbf{y}_v\right). \tag{31}$$

The loss of training $\mathrm{f}_{out}(\mathbf{x}_w; \boldsymbol{\theta})$ with pairs of feature and pseudo-label $(\mathbf{x}_w, y_v)$ of all nodes of $\mathcal{G}_v^{sub}$ is:

$$\mathcal{L}_{\mathcal{G}_v^{sub}} = \sum_{w \in \mathbb{V}_v^{sub}} \mathrm{CE}\left(\mathrm{f}_{out}(\mathbf{x}_w; \boldsymbol{\theta}), \mathbf{y}_v\right) \tag{32}$$

$$\tag{33}$$

Then, the corresponding gradient of $\mathcal{L}_{\mathcal{G}_v^{sub}}$ is :

$$\nabla_{\boldsymbol{\theta}} \mathcal{L}_{\mathcal{G}_v^{sub}} = \sum_{w \in \mathbb{V}_v^{sub}} \nabla_{\boldsymbol{\theta}} \mathrm{CE}\left(\mathrm{f}_{out}(\mathbf{x}_w; \boldsymbol{\theta}), \mathbf{y}_v\right). \tag{34}$$

By comparing Equation 31 and 34, we can see that training PDGNNs with a topology-aware embedding $\mathbf{e}_v$ equals to training the function $\mathrm{f}_{out}(\cdot; \boldsymbol{\theta})$ on all nodes of the computation ego-subgraph $\mathcal{G}_v^{sub}$ with a weight $\frac{\mathrm{f}_{out}(\mathbf{x}_w; \boldsymbol{\theta}) \cdot \pi(v, w; \hat{\mathbf{A}})}{\sum_{w \in \mathbb{V}_v^{sub}} \mathrm{f}_{out}\left(\mathbf{x}_w \cdot \pi(v, w; \hat{\mathbf{A}}); \boldsymbol{\theta}\right)}$ on each node to rescale the contribution dynamically.

$\square$

## A.3 FURTHER DISCUSSION ON PSEUDO-TRAINING EFFECTS OF GENERALIZED TE GENERATION FUNCTION

In this subsection, we give further analysis on the pseudo training effect when the TE generation follows the following formulation:

$$\mathbf{e}_v = \mathbf{g}(\{\mathbf{x}_w \mid w \in \mathbb{V}\}, \hat{\mathbf{A}}). \tag{35}$$

In this scenario, the pseudo training effect will depend on the specific form of $\mathbf{g}(\cdot, \cdot)$. Despite this, we can still analyze the strength of pseudo training effect with respect to the smoothness of the

function and the dataset properties. First of all, the pseudo training effect exists because the GNN models generate the prediction based on a local neighborhood. Therefore, the nodes with overlapping neighborhood (similar inputs to the model) share similar prediction results. If their labels are shared, then training these nodes could mutually reinforce each other. Accordingly, given an arbitrary function $\mathbf{g}(\cdot, \cdot)$, we can gain an insight into the strength of pseudo training effect by analyzing the similarity of the inputs when generating representations of different nodes. Without loss of generality, we assume $\mathbf{g}(\cdot, \cdot)$ be a continuous function (since $\mathbf{g}(\cdot, \cdot)$ does not require training, it does not have to be differentiable). Then, given two nodes $v$ and $w$, we denote their corresponding inputs to the model as two vectors $I_v$ and $I_w$. $I_v$ and $I_w$ may contain different neighborhood information based on the specific form of $\mathbf{g}(\cdot, \cdot)$. Now, it is obvious that the closer $I_v$ and $I_w$ are, the closer $\mathbf{g}(I_v, \hat{\mathbf{A}})$ and $\mathbf{g}(I_w, \hat{\mathbf{A}})$ are (due to the continuity of $\mathbf{g}(\cdot, \cdot)$). In other words, stronger homophily will lead to stronger pseudo training effect as we analyzed in Theorem 1 in the paper. Besides, the frequency components (in terms of the spectrum of the function, *e.g.*, with Fourier analysis) of $\mathbf{g}(\cdot, \cdot)$ also matters. If $\mathbf{g}(\cdot, \cdot)$ is mainly composed of low frequencies, *i.e.*, the change of $\mathbf{g}(\cdot, \cdot)$ is slow with respect to the change of the input, then the pseudo training effect is stronger because more nodes are getting similar representations. But if the function $\mathbf{g}(\cdot, \cdot)$ contains strong high frequency components, *i.e.* $\mathbf{g}(\cdot, \cdot)$ changes significantly with the change of input, then the pseudo training effect is weaker since only very similar inputs of the nodes get similar outputs.

In experiments, we also instantiated $\mathbf{g}(\cdot, \cdot)$ with the reservoir computing module (Gallicchio & Micheli, 2020), which yields comparable performance with other instantiations (Section B.2).

### A.4 PSEUDO-TRAINING EFFECT AND GRAPH HOMOPHILY

First of all, we give the formal definition of the graph homophily ratio. Given a graph $\mathcal{G}$, the homophily ratio is defined as the ratio of the number of edges connecting nodes with a same label and the total number of edges, i.e.

$$h(\mathcal{G}) = \frac{1}{|\mathcal{E}|} \sum_{(j,k) \in \mathcal{E}} \mathbf{1}(\mathbf{y}_j = \mathbf{y}_k), \tag{36}$$

where $\mathcal{E}$ is the edge set containing all edges, $\mathbf{y}_j$ is the label of node $j$, and $\mathbf{1}(\cdot)$ is the indicator function (Ma et al., 2021). For any graph, the homophily ratio is between 0 and 1. For each computation ego-subgraph, when the homophily ratio is high, the neighboring nodes tend to share labels with the center node, and the pseudo training would be beneficial for the performance. Many real-world graphs like the social network and citation networks tend to have high homophily ratios, and pseudo training will bring much benefit, which is shown in Section 4.3 of the paper.

In our work, homophily ratio of the 4 graph datasets are: CoraFull-CL (0.567), Arxiv-CL (0.655), OGB-Products (0.807), Reddit-CL (0.755). These datasets cover the ones with high homophily (OGB-Products and Reddit), as well as the ones with lower homophily. The strong experiment results demonstrate that our proposed method performs well in all these scenarios.

Learning on more heterophilous graphs (homophily ratio close to 0) is also a promising direction, which requires $f_{topo}(\cdot)$ to be properly constructed, and is targeted by our future work.

Heterophilous graph learning is largely different from homophilous graph learning, and requires different GNN designs (Zheng et al., 2022). Therefore, for learning on heterophilous graphs, the function $f_{topo}(\cdot)$ of PDGNNs should also be instantiated to be suitable for heterophilous graphs. The rationale here is same as that the classic GNNs (MPNNs) for homophilous graphs perform badly on heterophilous graphs. And the heterophilous GNNs require essentially different model structures (Zheng et al., 2022; Abu-El-Haija et al., 2019; Zhu et al., 2020).

In the following, we will first explain how to configure $f_{topo}(\cdot)$ for heterophilous graphs, and then explain why it ensures that pseudo-training will not damage the performance.

The key difference of heterophilous graph learning is that the nodes belonging to the same classes are not likely to be connected, and GNNs should be designed to separately process the proximal neighbors with similar information and distal neighbors with irrelevant information, or only aggregate information from the proximal neighbors (Zheng et al., 2022; Abu-El-Haija et al., 2019; Zhu et al., 2020).

Accordingly, the first strategy to construct $f_{topo}(\cdot)$ is to follow the MixHop (Abu-El-Haija et al., 2019) and let $f_{topo}(\cdot)$ encodes neighbors from different hops separately. Specifically, a given computation ego-subgraph (the input to $f_{topo}(\cdot)$) will be divided into different hops. For each hop, $f_{topo}(\cdot)$ generates a separate embedding. Finally, the embeddings of different hops are concatenated (summation should not be used to ensure different hops are separately processed) as the final TE (Equation 4 in the paper).

The second strategy follows H2GCN (Zhu et al., 2020) to only aggregate higher-order neighbors, because H2GCN (Zhu et al., 2020) theoretically justifies that two-hop neighbors tend to be proximal to the center node, if the the one-hop neighbors have labels that are conditionally independent of the center node's label. In other words, for designing $f_{topo}(\cdot)$, the one-hop neighbors can be simply ignored when doing neighborhood aggregation.

In other words, via constructing $f_{topo}(\cdot)$ to be suitable for heterophilous graphs, the neighborhood aggregation is still conducted on the proximal nodes, and so is the pseudo-training. In this way, the pseudo-training will not damage but still benefit the performance.

Incorporating heterophilous graphs into continual learning on expanding graphs is promising and interesting. In our future works, we will construct continual learning tasks on heterophilous graphs and implement suitable models as introduced above.

## B  ADDITIONAL EXPERIMENTAL RESULTS & SETUPS

In this section, we provide additional information on the datasets, experimental settings, and experimental results.

### B.1  VISUALIZATION OF NODE EMBEDDINGS WITH A TASK SEQUENCE

To interpret the learning process of PDGNNs-TEM, we visualize the node embeddings of different classes with t-SNE (Van der Maaten & Hinton, 2008) while learning on a task sequence of 20 tasks over the Reddit dataset. In Figure 5, besides PDGNNs-TEM that replay data with topological information, we also show two other representative baselines, including ER-GNN for demonstrating how would the lack of topological information influence the performance, and Fine-tune for demonstrating the results without any continual learning technique. As shown in Figure 5, PDGNNs-TEM can ensure that the nodes from different classes are well separated while continuously learning new tasks sequentially (each color corresponds to a class). In contrast, for ER-GNN and Fine-tune, the boundaries of different classes are less clear, especially when more tasks are learnt.

### B.2  ADDITIONAL RESULTS OF STUDIES ON THE BUFFER SIZE

In this subsection, we show the performance of PDGNNs-TEM with different buffer sizes on the other 3 datasets in Figure 9 and 10. We observe similar patterns in these results, *i.e.*, the performance (both average accuracy and average forgetting) increases when the buffer size (in terms of the ratio of data) increases. Specifically, on OGB-Products dataset, which is the largest dataset with millions of nodes, the PDGNNs-TEM can achieve reasonably well performance with a buffer size of only 0.01 to the size of the dataset, which further demonstrates the effectiveness and efficiency of PDGNNs-TEM.

In Table 2 of the paper, we have the following findings: (1) our coverage maximization sampling does guarantee a superior coverage ratio compared to the other sampling strategies, especially when the buffer size is relatively small; (2) the performance does exhibit strong correlation with the coverage ratio, especially when the buffer size is small. For different buffer sizes, a higher coverage ratio can yield better performance. The performance gap between different sampling strategies is larger with smaller buffer sizes, which is also the situation when the coverage ratio gap is larger. In this case (buffer size smaller than 1.0%), the number of stored TEs is relatively small compared to the size of the dataset, therefore the effectiveness of pseudo training on more nodes is more prominent. With larger buffer sizes, all sampling strategies can cover a large ratio of nodes and the performance gaps close up. In real world applications, a smaller buffer size is typically adopted, therefore the high memory efficiency of coverage maximization sampling would be preferred.

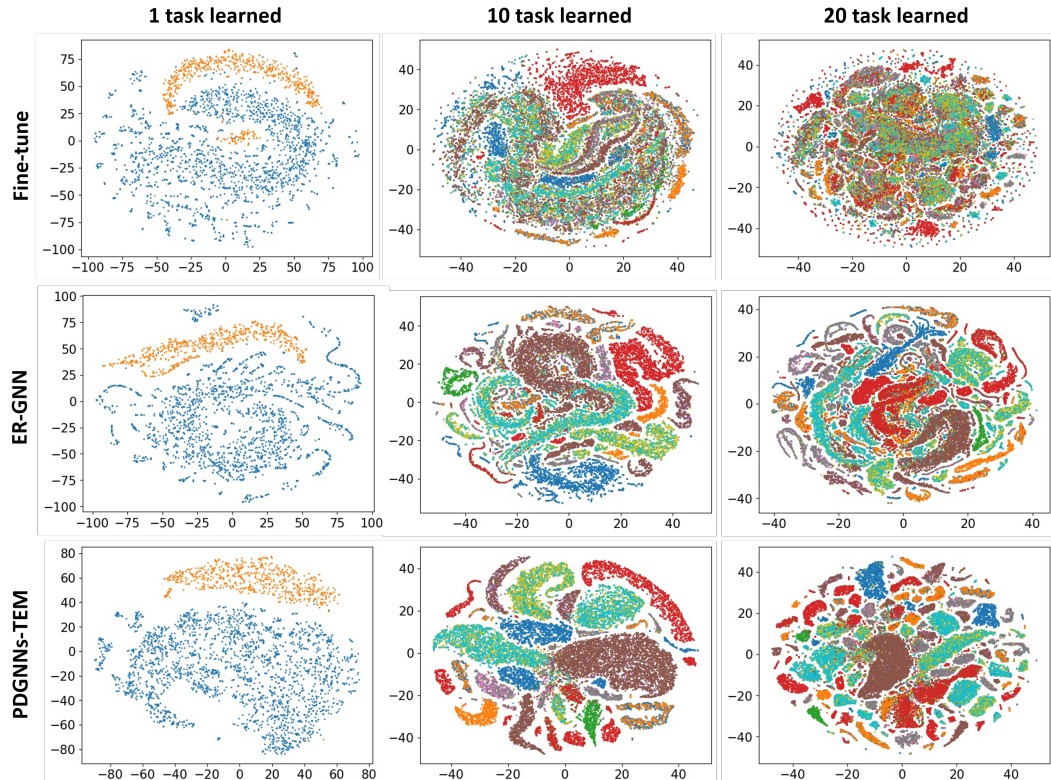

Figure 5: Visualization of node embeddings of different classes on Reddit dataset. The node representations are taken after learning 1, 10, and 20 tasks. From the top to the bottom, we show the results of Fine-tune, ER-GNN, and PDGNNs-TEM. Each color corresponds to a class.

Table 5: Performance comparisons under class-IL on OGB-Arxiv dataset with different task splittings (↑ higher means better).

| C.L.T. | 20 tasks | | 8 tasks | | 4 tasks | | 2 tasks | |
|---|---|---|---|---|---|---|---|---|
| | AA/% ↑ | FM/%↑ | AM/% ↑ | FM /% ↑ | AM/% ↑ | FM /% ↑ | AM/% ↑ | FM /% ↑ |
| Fine-tune | 4.9±0.0 | -89.7±0.4 | 10.5±0.1 | -77.5±0.5 | 16.4±0.2 | -63.9±0.6 | 26.4±0.3 | -47.3±0.9 |
| EWC 2017 | 8.5±1.0 | -69.5±8.0 | 9.4±0.1 | -73.7±1.1 | 15.7±0.3 | -62.8±0.7 | 24.8±0.3 | -47.5±0.6 |
| MAS 2018 | 4.8±0.4 | -72.2±4.1 | 10.3±0.2 | -77.5±0.6 | 16.5±0.3 | -64.0±0.5 | 26.3±0.6 | -47.5±0.7 |
| GEM 2017 | 4.9±0.0 | -89.8±0.3 | 10.7±0.1 | -81.5±0.3 | 18.2±0.2 | -70.6±0.5 | 31.3±0.1 | -58.5±0.2 |
| TWP 2021 | 6.7±1.5 | -50.6±13.2 | 8.3±0.4 | -66.1±1.3 | 14.0±0.4 | -57.6±1.5 | 22.0±0.4 | -47.6±0.5 |
| LwF 2017 | 9.9±12.1 | -43.6±11.9 | 24.2±0.4 | -31.9±1.0 | 19.6±1.1 | -41.8±1.7 | 19.6±0.7 | -51.1±0.1 |
| ER-GNN 2021 | 12.3±3.9 | -79.9±4.1 | 10.9±0.2 | -77.5±0.5 | 19.8±1.2 | -59.9±1.3 | 31.6±0.6 | -34.8±1.3 |
| Joint | 56.8±0.0 | -8.6±0.0 | 55.3±0.0 | -10.1±0.0 | 53.9±0.0 | -9.1±0.1 | 51.6±0.1 | -8.2±0.2 |
| PDGNNs* | 26.8±1.8 | -61.6±2.0 | 27.9±1.8 | -58.2±2.7 | 30.9±1.1 | -51.2±1.6 | 35.9±1.4 | -46.4±3.2 |
| PDGNNs | 53.2±0.4 | -14.7±0.4 | 51.6±0.4 | -15.0±0.7 | 50.6±0.4 | -12.8±0.5 | 49.7±0.3 | -11.4±0.5 |

The above analysis verifies our Theorem 2 and indicates higher coverage ratio would be beneficial to the performance.

Besides, in Figure 6, based on the class-IL scenario, we study the performance of PDGNNs-TEM on the OBG-Arxiv dataset when the buffer size (*i.e.*, the ratio of dataset) varies from 0.0002 to 0.6. Figure 6 exhibits the similar performance of different TE generation modules. Besides, when the buffer size grows from 0.0002 to 0.01, both the average accuracy and average forgetting of PDGNNs increase. When the buffer size reaches 0.1, the performance of PDGNNs is comparable to the setting which stores the entire training set (when the ratio of dataset is 0.6). These results demonstrate the efficiency of TEM. Moreover, the results in Figure 6 also show that the performance difference among different TE generation strategies is not significant.

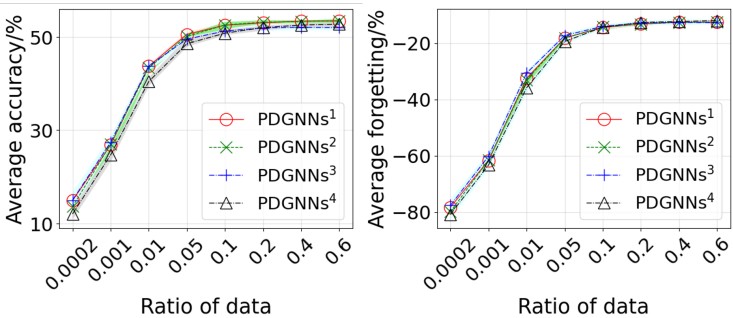

Figure 6: Average accuracy (left) and average forgetting (right) vs. buffer size on OGB-Arxiv. PDGNN[1] to PDGNN[3] instantiated $\pi(v, w; \hat{\mathbf{A}})$ as the forms introduced in Section 3.4 of the paper. PDGNN[4] adopts the reservoir computing module proposed in (Gallicchio & Micheli, 2020)

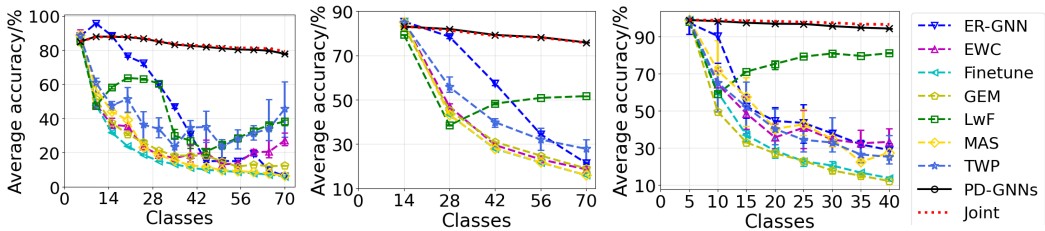

Figure 7: Dynamics of average accuracy on **CoraFull** dataset with task sequence of length of 14 (left), and 5 (middle) in class-IL scenario. Dynamics of average accuracy on **Reddit** dataset with task sequence of length of 8 (right) in class-IL scenario

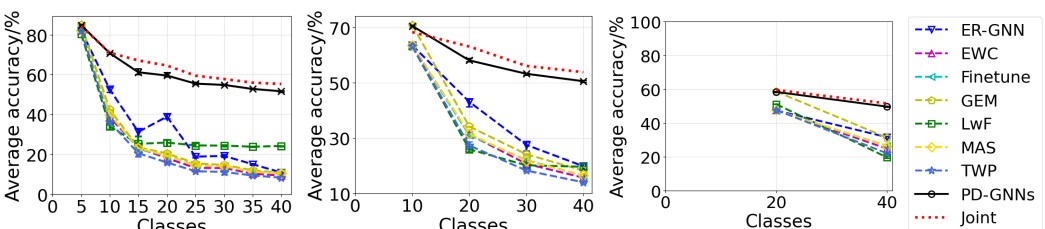

Figure 8: Dynamics of average accuracy on **OGB-Arxiv** dataset with task sequence of length of 8 (left), 4 (middle), and 2 (right) in class-IL scenario.

| C.L.T. | CoraFull | OGB-Arxiv | Reddit | OGB-Products |
|---|---|---|---|---|
| Full Subgraph | 7,264M | 35M | 2,184,957M | 5,341M |
| GEM | 7,840M | 86M | 329M | 82M |
| ER-GNN | 61M | 2M | 12M | 3M |
| SSM | 732M | 41M | 193M | 37M |
| PDGNNs-TEM | 37M | 2M | 9M | 2M |

Table 6: Memory space consumption of different methods.

### B.3 ADDITIONAL RESULTS OF COMPARISONS WITH THE STATE-OF-THE-ARTS

In this subsection, we provide additional results to compare PDGNNs-TEM with the baselines. In Table 5, we provide numerical results to compare different models and complement the curves of average accuracy provided in the paper. We list both the final average accuracy and average forgetting of all models on the OGB-Arxiv dataset with different task splittings in class-IL scenario. Besides, we also show the results of PDGNNs-TEM with an extremely small buffer size (*i.e.*, 0.001 of the size of the dataset), which is denoted with PDGNNs*. 0.001 of the size of OGB-Arxiv corresponds to storing only 4 examples per class and a total of 160 for 40 different classes, which is orders of magnitudes smaller than the buffer size of the memory based baselines with budgets of several hundred per class. From Table 5, we can observe that both PDGNNs and PDGNNs* significantly outperform the baselines. Even the PDGNNs* can outperform baselines by a large margin, which demonstrates the high efficiency of TEM. Considering that OGB-Arxiv contains 169,343 nodes, the performance of PDGNNs* is indeed impressive.

For clarification, we also provide analysis on why the joint training, the performance upper bound, could be outperformed by PDGNNs-TEM. First, the reason Joint (joint training) is regarded as the upper bound is because it learns all tasks simultaneously without the forgetting problem. However, the forgetting problem is not the only factor determining the performance. Therefore, when the forgetting problem is well addressed (like our proposed PDGNNs-TEM), the influence of other factors would emerge. Specifically,

1. PDGNNs-TEM learn tasks sequentially, the learning starts from few classes and gradually learn more new classes in the new tasks. In contrast, Joint learns all classes (tasks) simultaneously. These two learning manners result in different optimization difficulties. For datasets whose task sequence happen to contain easy-to-hard sub-sequences, learning the tasks sequentially is easier than learning jointly. This phenomenon is also studied in existing works (Bhat et al., 2021). Therefore, when the forgetting problem is well addressed and is not the dominating factor, PDGNNs may outperform joint training when sequential learning is more suitable for the given dataset.

2. When learning a task in the task sequence, information of the previous tasks is provided to PDGNNs through the stored TEs, which are representative data selected from the original data. While for Joint, all of the original data are used, which may contain noisy data that are detrimental to the performance in sometime. In other words, noise may be reduced by our proposed coverage max sampling, and replaying the selected TEs is better than training with the complete original data.

Besides the results obtained on the longest task sequences shown in the paper, in this section, we also show the results with different task splittings (with class-IL scenario) in Figure 7 and 8. Note that the task sequence of length is equivalent to the number of tasks to learn (as shown in Table 8) for each dataset.

### B.4 COMPARISON ON THE SPACE COMPLEXITY

To concretely demonstrate the memory efficiency of the proposed PDGNNs-TEM, we a comparison on the space usage of different memory based methods. The final memory consumption (measured with number of float32 parameters) after learning each entire dataset are shown below. As a reference, the memory of storing full computation ego-subgraph is also calculated.

From the table above, it is obvious that our proposed PDGNNs-TEM is indeed highly efficient in terms of the memory space usage.

| | #layers | Fine-tune | PDGNNs-TEM | Joint |
|---|---|---|---|---|
| **1** | AA/% ↑ | $4.9 \pm 0.0$ | $49.9 \pm 0.0$ | $48.2 \pm 0.3$ |
| | AF/% ↑ | $-89.9 \pm 0.2$ | $0.7 \pm 0.5$ | $-15.9 \pm 0.5$ |
| **2** | AA/% ↑ | $4.9 \pm 0.0$ | $53.2 \pm 0.2$ | $56.8 \pm 0.0$ |
| | AF/% ↑ | $-89.7 \pm 0.4$ | $-14.7 \pm 0.2$ | $-8.6 \pm 0.4$ |
| **3** | AA/% ↑ | $4.8 \pm 0.0$ | $48.5 \pm 0.6$ | $31.1 \pm 0.7$ |
| | AF/% ↑ | $-75.3 \pm 7.1$ | $7.1 \pm 0.2$ | $-38.8 \pm 7.9$ |
| **4** | AA/% ↑ | $3.5 \pm 0.9$ | $38.7 \pm 1.2$ | $9.8 \pm 1.2$ |
| | AF/% ↑ | $-68.7 \pm 4.7$ | $-13.1 \pm 0.9$ | $-39.7 \pm 0.4$ |
| **5** | AA/% ↑ | $2.5 \pm 0.0$ | $2.5 \pm 0.0$ | $6.3 \pm 0.7$ |
| | AF/% ↑ | $-29.9 \pm 11.2$ | $-29.1 \pm 0.5$ | $-29.1 \pm 0.8$ |

Table 7: Caption

Table 8: The detailed statistics of datasets and task splittings

| **Dataset** | CoraFull 2000 | OGB-Arxiv 2020[2] | Reddit 2017 | OGB-Products 2020[3] |
|---|---|---|---|---|
| # nodes | 19,793 | 169,343 | 232,965 | 2,449,029 |
| # edges | 130,622 | 1,166,243 | 114,615,892 | 61,859,140 |
| # classes | 70 | 40 | 40 | 47 |
| # tasks | 35 / 14 / 5 / 2 | 20 / 8 / 5 / 2 | 20 / 8 / 5 / 2 | 23 / 10 / 5 / 3 |

Note: The memory space efficiency comparison can only be made among the memory-replay based methods, since the other methods do not store observed data. But this is not a disadvantage of memory replay based methods. Instead, thanks to the memory buffer, memory based methods are currently the most effective continual learning techniques, which significantly outperform the other methods without memory, as shown in Table 3 of the paper.

## B.5 INVESTIGATION ON THE MODEL DEPTH

In this subsection, we demonstrate how the model performance changes with the model depth. The experiments are conducted on OGB-Arxiv, and the model depth varies from 1 layer to 5 layers.

As shown in Table 7, the performance is good when the model is not very deep. When the depth increases to 3-layer, all methods, exhibit performance decrease. In this stage, PDGNNs-TEM show significant superiority over Joint. The potential reason is that Joint uses full historical data, and is more prone to over-fitting. In contrast, PDGNNs-TEM selectively stores only selecting a subset of historical data, therefore could alleviate over-fitting. When the depth further increases, the performance of all methods keeps going down.

The influence of the model depth on the performance is actually determined by the design of the backbone GNNs, which is off the scope of our work. In our experiments, we adopt the best performing configuration with 2 layers.

## B.6 DATASET DESCRIPTIONS

The statistics of the datasets are summarized in Table 8. Among these datasets, CoraFull and OGB-Arxiv are two citation graphs, Reddit is a graph constructed from Reddit posts, and OGB-Products is an Amazon product co-purchasing network. The usage of the datasets is granted for academic purposes, and full details on the licenses can be obtained from the official websites. The datasets contain no personally identifiable information or offensive content.

---

[2] https://ogb.stanford.edu/docs/nodeprop/#ogbn-arxiv
[3] https://ogb.stanford.edu/docs/nodeprop/#ogbn-products

### B.6.1 CITATION NETWORKS

CoraFull (McCallum et al., 2000) is a citation network labeled based on the paper topics. In total, it contains 19,793 nodes and 126,842 edges. The original dataset has 65,311 edges. We directly adopted the version in DGL with reverse edges added and duplicates removed. It contains 70 classes, and each node has a 8,710 dimensional feature vector.

The OGB-Arxiv dataset is collected in the Open Graph Benchmark (Hu et al., 2020) OGB. It is a directed citation network between all Computer Science (CS) arXiv papers indexed by MAG (Wang et al., 2020d). Totally it contains 169,343 nodes and 1,166,243 edges. The dataset contains 40 classes.

### B.6.2 SOCIAL NETWORK

Reddit (Hamilton et al., 2017) is a graph dataset from Reddit posts made in the month of September, 2014. The node labels are the community, or "subreddit", that the posts belong to. The authors sampled 50 large communities and built a post-to-post graph, connecting posts if the same user comments on both. In total this dataset contains 232,965 nodes with an average degree of 492, 114,615,892 edges, and a 602 dimensional feature vector for each node. We directly used the version integrated in DGL library.

### B.6.3 PRODUCT CO-PURCHASING NETWORK

OGB-Products is collected in the Open Graph Benchmark (Hu et al., 2020) [4], representing an Amazon product co-purchasing network [5]. It contains 2,449,029 nodes and 61,859,140 edges. Nodes represent products sold on Amazon, and edges indicate that the connected products are purchased together. In our experiments, we select 46 classes and omit the last class containing only 1 example.

### B.7 ADDITIONAL DETAILS ON EXPERIMENT SETUP AND MODEL EVALUATION

**Continual learning setting**. In this part, we give concrete examples to further explain the difference between class-IL and task-IL scenarios. In class-IL scenario, a model has to classify the given data by picking a class from all previously learnt classes, while the task-IL scenario only require the model to distinguish the classes within each task. Concretely, suppose the model learns on a citation network with a two-class task sequence {(*physics*, *chemistry*), (*biology*, *math*)}. In class-IL scenario, after training, the model is required to classify a given document into one of the four classes. In task-IL scenario, the model is only required to classify a given document into to (*physics*, *chemistry*) or (*biology*, *math*), while cannot distinguish between *physics* and *biology* or between *chemistry* and *math*.

For each dataset, the splitting of different tasks is conducted by dividing the classes into groups in the default order. Different group sizes are shown in Table 1 of the paper. For each task, the ratio for training, validation, and testing is 60%, 20%, 20%. The validation set was only used in baseline model selection, since the hyperparameters of our method are simply set to be consistent with baselines (Section 4.2 in the paper). For all baselines and our method, the number of training epochs is 200. The two large datasets Reddit and OGB-Products requrie mini-batch training, and the batch size is chosen as 2,000.

**Baselines and model settings**. In this part, we give more details on the model configurations. The following setting applies to all datasets. All the backbone GNNs of baselines are configured as 2-layer with 256 hidden dimensions, which exhibit better performance than other configurations. To ensure a fair comparison, we also set the MLP part of PDGNNs as 2-layer with 256 hidden dimensions (the TE generation part does not contain trainable parameters) as shown in Table 9. The memory budget (number of nodes per class selected to store) is set as 400 for PDGNNs-TEM for all datasets. For the memory based baselines, the budget was chosen with two criteria: 1. The buffer size should be large than the size for PDGNNs-TEM to ensure PDGNNs-TEM does not succeed by storing more examples. 2. The budget should be large enough for the baseline methods to gain a reasonable

---

[4] https://ogb.stanford.edu/docs/nodeprop/ogbn-products
[5] http://manikvarma.org/downloads/XC/XMLRepository.html

Table 9: The configuration of the MLP part of PDGNNs.

| No. layer | Input dimensions | Output dimensions | Activation |
|:---:|:---:|:---:|:---:|
| 1 | # data dimensions | 256 | ReLU |
| 2 | 256 | # classes | SoftMax |

Table 10: Memory budget of different methods on different datasets.

| | CoraFull | OGB-Arxiv | Reddit | OGB-Products |
|:---|:---:|:---:|:---:|:---:|
| PDGNNs-TEM (allocated) | 400 | 400 | 400 | 400 |
| PDGNNs-TEM (actual consumption per class) | 165 | 361 | 395 | 315 |
| Baselines (allocated) | 500 | 500 | 600 | 700 |
| Baselines (actual consumption per class) | 169 | 431 | 590 | 507 |

performance. The allocated budgets and the actual memory consumption [6] on different datasets are listed in Table 10, which demonstrates that PDGNNs-TEM is actually highly efficient in using the buffered data and outperforms the memory based baselines with less memory usage. The reason that the allocated budgets are different from the actual memory consumption is that some classes are smaller than the allocated budget. A brief introduction of the baseline continual learning techniques are given below:

1. **Fine-tune** directly trains a given backbone GNN on the task sequence without any technique to avoid forgetting, therefore can be viewed as a lower bound on the continual learning performance.

2. **Elastic Weight Consolidation (EWC) (Kirkpatrick et al., 2017)** adds a quadratic penalty to prevent the model weights, which are important to prevent model parameters related to previous tasks from shifting too much.

3. **Memory Aware Synapses (MAS) (Aljundi et al., 2018)** measures the importance of the parameters according to the sensitivity of the predictions on the parameters and slows down the update of the important parameters.

4. **Gradient Episodic Memory (GEM) (Lopez-Paz & Ranzato, 2017)** stores representative data in episodic memory and adds a constraint to prevent the loss of the episodic memory from increasing and only allow it to decrease.

5. **Topology-aware Weight Preserving (TWP) (Liu et al., 2021)** adds a penalty on the model weights to preserve the topological information of previous graphs.

6. **Learning without Forgetting (LwF) (Li & Hoiem, 2017)** uses knowledge distillation to constrain the shift of parameters for old tasks.

7. **Experience Replay GNN (ER-GNN) (Zhou & Cao, 2021)** integrates memory-replay to GNNs by storing experience nodes from previous tasks.

8. **Sparsified Subgraph Memory (SSM)** (Zhang et al., 2022d) stores the sparsified version of the representative computation ego-subgraphs for memory replay.

9. **Joint Training** does not follow the continual learning setting and trains the model on all tasks simultaneously. Therefore, Joint Training does not suffer from forgetting problems and its performance can be viewed as the upper bound for continual learning.

A widely adopted performance upper bound on the continual learning models is joint training. Different from being trained sequentially on a task sequence, a jointly trained model does follow the continual learning setting but is simultaneously trained on all tasks. Therefore, jointly trained models do not suffer from the forgetting problem and could be viewed as an upper bound on the continual learning performance. Note that under the class-IL setting, the average accuracy of the jointly trained

---

[6]Since the sizes of different classes differ significantly, classes whose sizes are smaller than the budget do not consume memory to the allocated budget.

model will still decrease as the number of classes increases. The reason is that the classification difficulty increases when the number of classes vary from small to large.

**Model Evaluation**.

Evaluating a model under continual learning setting is more complex than that under the standard setting, which only requires a single numerical value (accuracy, F1 score, *etc.*). Specifically, after learning each new task, the performance of a model on all previous tasks would change. Therefore, the most thorough evaluation would be showing the performance on all previous tasks after learning each new task. To this end, we denote the performance of a model trained consecutively on $T$ tasks with an accuracy matrix $A \in \mathbb{R}^{T \times T}$, in which each entry $A_{i,j}$ denotes the model's accuracy on task $j$ after learning task $i$. Then each row $A_{i,:}$ of $A$ show the model's accuracy on all previous tasks after learning task $i$ and each column $A_{:,j}$ shows how the model's accuracy on task $j$ changes when being trained sequentially on all the tasks. Besides the accuracy matrix, to more concretely show the dynamics of the overall performance, we also adopt the curve of average accuracy as a tool. To plot this curve, we compute the average accuracy after learning each task, *i.e.*:

$$\left\{ \frac{\sum_{j=1}^{i} A_{i,j}}{i} | i = 1, ..., T \right\}. \tag{37}$$

Denoting the forgetting on task $j$ after learning task $i$ as $A_{i,j} - A_{j,j}$, the dynamics of the average forgetting is similarly defined as:

$$\left\{ \frac{\sum_{j=2}^{i} A_{i,j} - A_{i,j-1}}{i-1} | i = 1, ..., T \right\}. \tag{38}$$

When a single numerical value is preferred to evaluate the performance, the final average accuracy and average forgetting after learning all $T$ tasks could be used.

**Class imbalance in continual learning on expanding graphs**. According to Equation 39, the performance on different tasks contributes equally to the average accuracy. However, unlike the traditional continual learning with balanced datasets, the class imbalance problem is usually severe in graphs, of which the effect will be entangled with the effect of forgetting. Directly balancing the data by choosing equal number of nodes from each class may not be practical. For example, in the OGB-Products dataset, the largest class has 668,950 nodes, while the smallest contains only 1 node. Therefore, sampling equal amount of nodes from each class would result in either deleting many classes without enough nodes or sampling a very small number of nodes from each class so that all classes can provide the same amount of nodes. Moreover, deleting nodes in a graph would also change the original topological structures of the remaining nodes, which is undesired.

To this end, we propose to re-scale the loss of nodes in each class according to the class sizes. Denoting the set of the classes of our training data as $\mathcal{C}$, the number of examples of each class in $\mathcal{C}$ can be represented as $\{n_c \mid c \in \mathcal{C}\}$. Then, we calculate a scale for each class $c$ to balance their contribution in the loss function as $s_{\mathbf{y}_v} = \frac{n_c}{\sum_{i \in \mathcal{C}} n_i}$, where $\mathbf{y}_{v,c} = 1$. Finally, our balanced loss is:

$$\mathcal{L} = \sum_{v \in \mathbb{V}_\tau} l(f(\mathbf{e}_v; \boldsymbol{\theta}), \mathbf{y}_v) \cdot s_{\mathbf{y}_v} + \sum_{\mathbf{e}_w \in \mathcal{TEM}} l(f(\mathbf{e}_w; \boldsymbol{\theta}), \mathbf{y}_w) \cdot s_{\mathbf{y}_w}. \tag{39}$$

Since the evaluation treats all classes equally and the loss on each class is balanced, $\lambda$ is omitted in our implementation, as it will influence the balance of each class.

**Class-incremental classifier**. In standard classification tasks, the number of the output heads of a model equals the number of classes and is fixed at the beginning. But in class-IL setting, the output heads will continually increase along with the new classes. To better accommodate new classes, cosine distance is adopted by several works (Wu et al., 2021; Wang et al., 2018; Gidaris & Komodakis, 2018) to slightly modify the standard softmax classifier. Empirically, PDGNNs with TEM outperform the standard softmax classifier which simply increases the output heads with the number of classes. All baselines are tested with both strategies and the one that achieves better performance over the validation set is employed for comparison. Specifically, only LwF exhibits better performance with the cosine distance based classifier.

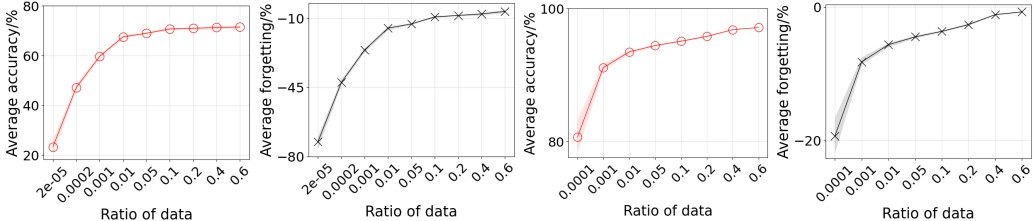

Figure 9: Average accuracy (Red circles) and average forgetting (Black crosses) changes with buffer size on OGB-Products dataset (the left two) and Reddit dataset (the right two).

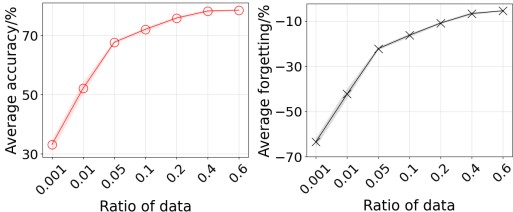

Figure 10: Average accuracy (Red circles) and average forgetting (Black crosses) changes with buffer size on CoraFull dataset.

## C  ADDITIONAL DETAILS OF PARAMETER DECOUPLED GNNS WITH TEM

### C.1  MEMORY UPDATE

As mentioned in Section 3.3 of the paper, in real-world applications, the data may come in batches instead of being presented simultaneously. Therefore, the updating of $\mathcal{TEM}$ may need modification. The key issue is to determine how to update $\mathcal{TEM}$ such that the newly sampled TEs can be accommodated accordingly. We present two different approaches to handle this.

1. The most straightforward approach is to store the computation ego-subgraph size $s_v^{sub}$ of each $\mathbf{e}_v$ and recalculate the multinomial distribution. Given the incoming new node set $\mathbb{V}_\tau$, the probability of sampling each node is recalculated as $p_v = \frac{s_v^{sub}}{\sum_{w \in \mathbb{V}_\tau \bigcup \mathcal{TEM}_i} s_w^{sub}}$. Then $n$ TEs are sampled to populate the $\mathcal{TEM}$.

2. For efficiency, we can also adopt the reservoir sampling based strategy to update existing TEs in $\mathcal{TEM}$ without recalculating the multinomial distribution. Specifically, given a new node set $\mathbb{V}_\tau$, we first sample $\min\{n, |\mathbb{V}_\tau|\}$ nodes (TEs) $\mathcal{S}$ from $\mathbb{V}_\tau$ with the coverage maximization sampling. Next, we align all TEs in $\mathcal{TEM}$ and $\mathcal{S}$ in a sequence, *i.e.* the first $|\mathcal{TEM}|$ elements are from $\mathcal{TEM}$ and the following elements are from $\mathcal{S}$. Finally, for each TE $\mathbf{e}_v$ in $\mathcal{S}$, suppose its order in the sequence is $o_v \in \{|\mathcal{TEM}|, |\mathcal{TEM}| + 1, ..., |\mathcal{TEM}| + |\mathcal{S}|\}$, we generate a random integer $r$ from uniform distribution on 1 to $|\mathcal{TEM}| + o_v\}$. If $r$ falls in the range from 1 to $|\mathcal{TEM}|$, then the $r$-th TEs in $\mathcal{TEM}$ is replaced by $\mathbf{e}_v$, otherwise $\mathbf{e}_v$ is deleted. In this way, the nodes in $\mathcal{TEM}$ can be randomly updated with the newly sampled TEs.

### C.2  DIFFERENTIATION BETWEEN COVERAGE MAXIMIZATION SAMPLING AND SEVERAL OTHER WORKS WITH SIMILAR NAMES

In this work, we propose the Coverage Maximization Sampling to maximize the number of nodes covered by the computation ego-subgraphs of the selected TEs. Several other works that may be confused with our proposed sampling method include the coverage maximization in ER-GNN (Zhou & Cao, 2021) and the influence maximization works on social networks (Chen et al., 2009; Wang et al., 2016).

First, although the names are same, our coverage maximization is entirely different from the one of ER-GNN. The one in ER-GNN follows the idea to maximally cover the node attribute/embedding space. Specifically, given the attributes/embeddings of a set of nodes, they aim to maximize the mutual distance among the selected subset of nodes to maximally cover the attribute/embedding space. In contrast, our coverage maximization aims to select the TEs of the nodes with larger computation ego-subgraphs, and does not consider the coverage in attribute/embedding space. Our design is based on our theoretical finding of the pseudo-training effect (Section 3.5,3.6) and is specially beneficial for continual learning.

Second, the influence maximization in (Chen et al., 2009; Wang et al., 2016) has a different target compared to our proposed coverage maximization sampling. Our maximization coverage sampling is based on our theoretical analysis on the pseudo-training effect. Specifically, we theoretically demonstrate that retraining the TE of a node, to some extent, is equivalent to training with all nodes in the computation ego-subgraph. Therefore, we aim to choose the nodes with larger computation ego-subgraphs. However, in an influence maximization problem, the influence of a node can only be disseminated to its neighbors with a specific propagation probability. Therefore, given the computation ego-subgraph of a node v, the influence of v only spreads to part of the computation ego-subgraph, which is inconsistent with the information propagation in GNNs, and does not faithfully reflect the number of nodes benefiting from the pseudo-training effect. Above all, coverage maximization is a natural choice given our theoretical analysis, while influence maximization serves for other purposes.

## C.3 Detailed Description of the Workflow of PDGNNs-TEM

---

**Algorithm 2** Training of PDGNNs-TEM

---

**Input:** Training task sequence $\mathcal{S} = \{\mathcal{G}_1, \mathcal{G}_2,...,\mathcal{G}_T\}$, function $f_{topo}(\cdot)$, $f_{out}(\cdot; \boldsymbol{\theta})$, sampling function sampler$(\cdot, \cdot)$, memory budget $n$, loss function $l(\cdot, \cdot)$, neural network optimizer Optim$(\cdot, \cdot)$, number of epochs $M$.

**Output:** Constructed memory buffer $\mathcal{TEM}$, trained parameters $\boldsymbol{\theta}$.

1: Initialize $\mathcal{TEM} = \{\}$.
2: **for each** $\mathcal{G} \in \mathcal{S}$ **do**
3:     **for each** epoch $\in \{1, ..., M\}$ **do**
4:         Calculate the TEs for nodes in node set $\mathbb{N}_\tau$ of $\mathcal{G}_\tau$
5:         $\{\mathbf{e}_v \mid \mathbf{e}_v = f_{topo}(\mathcal{G}_v^{sub}), v \in \mathbb{V}_\tau\}$
6:         Calculate final prediction of nodes in $\mathbb{N}_\tau$
7:         $\{\hat{\mathbf{y}}_v \mid \hat{\mathbf{y}}_v = f_{out}(\mathbf{e}_v; \boldsymbol{\theta}), v \in \mathbb{V}_\tau\}$
8:         Calculate loss of the current task $\tau$
9:         $\mathcal{L}_\tau = \sum_{v \in \mathbb{V}_\tau} l(f_{out}(\mathbf{e}_v; \boldsymbol{\theta}), \mathbf{y}_v)$
10:        Calculate the TEs for nodes in memory $\mathcal{TEM}$
11:        $\{\mathbf{e}_v \mid \mathbf{e}_v = f_{topo}(\mathcal{G}_v^{sub}), v \in \mathcal{TEM}\}$
12:        Calculate final prediction of nodes in $\mathcal{TEM}$
13:        $\{\hat{\mathbf{y}}_v \mid \hat{\mathbf{y}}_v = f_{out}(\mathbf{e}_v; \boldsymbol{\theta}), v \in \mathbb{V}_\tau\}$
14:        Calculate the auxiliary loss for preventing forgetting
15:        $\mathcal{L}_{aux} = \sum_{\mathbf{e}_w \in \mathcal{TEM}} l(f_{out}(\mathbf{e}_w; \boldsymbol{\theta}), \mathbf{y}_w)$
16:        Calculate balancing parameter $\lambda$ according to the class sizes in $\mathbb{N}_\tau$ and $\mathcal{TEM}$
17:        Calculate the total loss
18:        $\mathcal{L}_{total} = \mathcal{L}_\tau + \lambda \mathcal{L}_{aux}$
19:        Calculating the gradients of the trainable parameters $\frac{\partial \mathcal{L}_{total}}{\partial \boldsymbol{\theta}}$.
20:        Update $\boldsymbol{\theta}$ with the optimizer
21:        $\boldsymbol{\theta} = \text{Optim}(\boldsymbol{\theta}, \frac{\partial \mathcal{L}_{total}}{\partial \boldsymbol{\theta}})$
22:     **end for each**
23:     Update the memory buffer
24:     $\mathcal{TEM} = \mathcal{TEM} \bigcup \text{sampler}(\{\mathbf{e}_v \mid v \in \mathbb{V}_\tau\}, n)$
25: **end for each**

---

In this subsection, we provide an algorithm to describe the working procedure of PDGNNs-TEM step by step. The training process iterates over the tasks sequences containing $\mathcal{T}$ subgraphs.

During training, when learning on each task, function $f_{topo}(\cdot)$ will first generate the TE for each node, which is subsequently fed into the output function $f_{out}(\cdot; \boldsymbol{\theta})$ to get the final prediction. After obtaining the predictions, the loss $\mathcal{L}_\tau$ is calculated based on the predictions and the ground-truth labels. The same procedure is also conducted for data in the memory buffer $\mathcal{TEM}$ to get the loss

$\mathcal{L}_{aux}$. Then a total loss is obtained by summing up $\mathcal{L}_\tau$ and $\mathcal{L}_{aux}$ with $\lambda$ to balance the contribution from classes with different sizes. After learning each task, the memory buffer will be updated with new representative data selected by the sampling function $\mathrm{sampler}(\cdot, \cdot)$.

During testing, only the inference stage is conducted. The given data will sequentially go through function $\mathrm{f}_{topo}(\cdot)$ and $\mathrm{f}_{out}(\cdot; \boldsymbol{\theta})$ to get the final prediction.

---

**Algorithm 3** Inference of PDGNNs-TEM

---

    **Input:** Test task sequence $\mathcal{S} = \{\mathcal{G}_1, \mathcal{G}_2,...,\mathcal{G}_T\}$, function $\mathrm{f}_{topo}(\cdot)$, $\mathrm{f}_{out}(\cdot; \boldsymbol{\theta})$.
    **Output:** Constructed memory buffer $\mathcal{TEM}$, trained parameters $\boldsymbol{\theta}$.
1: **for each** $\mathcal{G} \in \mathcal{S}$ **do**
2:     Calculate the TEs for nodes in node set $\mathbb{N}_\tau$ of $\mathcal{G}_\tau$
3:     $\{\mathbf{e}_v \mid \mathbf{e}_v = \mathrm{f}_{topo}(\mathcal{G}_v^{sub}), v \in \mathbb{V}_\tau\}$
4:     Calculate final prediction of nodes in $\mathbb{N}_\tau$
5:     $\{\hat{\mathbf{y}}_v \mid \hat{\mathbf{y}}_v = \mathrm{f}_{out}(\mathbf{e}_v; \boldsymbol{\theta}), v \in \mathbb{V}_\tau\}$
6: **end for each**

---

## D   Remaining Challenges and Future Directions

Our proposed PDGNNs-TEM has successfully teckled the memory explosion problem in continual learning on expanding graphs, and brought the performance to a new level especially under the class-IL scenario. However, there are still remaining challenges in the field waiting to be overcome. A prominent challenge is that most methods for continual learning on expanding graphs, including our proposed PDGNNs-TEM, require clear task boundaries when learning on a sequence of tasks. But in real-world applications, the distribution of the data may shift gradually without explicit boundaries. In this case, for the regularization based methods (*e.g.* EWC, TWP), mechanisms are needed to decide when to record the importance of the parameters. For memory replay based methods (*e.g.* our proposed PDGNNs-TEM, SSM), how to ensure the data from different tasks are balanced in the buffer becomes challenging. For the parameter-isolation based methods (*e.g.* HPNs), the model would need to know when to allocate new parameters. Above all, significant opportunities for future exploration and inquiry persist within the domain of continual learning on expanding graphs, which will be investigated by our future works.

## E   Broader Impact

In this paper, we proposed a general technique to enable GNNs which can fit into the PDGNNs framework to continually learn on expanding networks. The method can be applied to any scenario requiring generating node representations on networks. The results of this paper can have an immediate and strong impact to address existing challenges for continual learning on expanding graphs, enabling to achieve state-of-the-art performance, and thus positively impacting applications on social networks, recommender systems, dynamic systems, *etc.*

Potential negative social impact may arise depending on the application scenario. For example, the privacy issue should be carefully considered when dealing with data containing user information.

