# OpenReview forum: "Topology-aware Embedding Memory for Learning on Expanding Graphs"
_ICLR.cc/2024/Conference — Submitted to ICLR 2024_

### Official Review · Reviewer_cucL · 2023-10-26

**Soundness:** 3 good
**Presentation:** 3 good
**Contribution:** 3 good
**Rating:** 6
**Confidence:** 4

**Summary:**

The paper proposes to use topology-aware embedding to preserve the subgraph information in the buffer for the continual learning. the methods can reduce the memory usage for the memory replay. Experiments show that the approach is promising

**Strengths:**

1. the paper is well written and easy to follow
2. the experiments look good

**Weaknesses:**

From my understanding,  the Topology-aware Embedding Memory is to replace the dynamic message passing aggregation of the subgraph with the fixed representation, I have concern about the novelty.

**Questions:**

how does the subgraph size influence the quality of the topology-aware Embedding? and how to choose the subgraph size for computing?

---

> ### Author Response · Authors · 2023-11-15
> **Responses to Reviewer cucL**
>
> We sincerely appreciate the reviewer for recognizing our writing quality and empirical merits. We have carefully prepared responses to the concerns/questions from the reviewer below.
>
> **Q1. From my understanding, the Topology-aware Embedding Memory is to replace the dynamic message passing aggregation of the subgraph with the fixed representation, I have concern about the novelty.**
>
> **A1.** Thanks for this nice concern, and we are happy to have the opportunity to further clarify our novelty.
>
> First, besides the TE generation (i.e. replacing the message passing with fixed representations mentioned by the reviewer), which is a part of our proposed PDGNNs-TEM framework, the whole framework is innovative, effective, and has important practical implications. Our proposed PDGNNs-TEM, for the first time, resolves the memory explosion problem and enable full preservation of the topological information. As recognized by the reviewer, this framework can significant improve the performance on multiple large datasets, especially in the challenging class-IL scenario. *This general framework for tackling the memory explosion problem was never developed in existing works*.
>
> Second, we made the first theoretical attempt to analyze the relationship between the stored data and the remaining ones in the context of continual learning on expanding graphs. This effort resulted in the theoretical results on the pseudo-training effect (**Section 3.5**).
>
> Third, based on the theoretical findings, we proposed the novel coverage maximization sampling technique (**Section 3.6**). This technique could enlarge the coverage ratio and further boost the performance, especially when the memory budget is tight.
>
> Above all, besides the operation to replace message passing, our work actually consists many innovative elements including the framework, theoretical analysis, and the coverage maximization sampling scheme.
>
> **Q2. how does the subgraph size influence the quality of the topology-aware Embedding? and how to choose the subgraph size for computing?**
>
> **A2.**
>
> Thanks for the nice question, which is a keen observation. In our work, the theoretical analysis on the pseudo-training effect (**Section 3.5**) can answer this question.
>
> According to the pseudo-training effect (**Section 3.5**), the larger the computation ego-subgraph, the more nodes benefit from the mempry replay. Therefore, the Topology-aware Embeddings (TEs) with larger computation ego-subgraphs are preferred. Accordingly, we have defined the concept of coverage ratio and developed the novel coverage maximization sampling strategy (**Section 3.6**) for obtaining higher quality TEs. The effectiveness of coverage maximization sampling is also validated in experiments (**Table 1 in page 7, Section 4.3**).
>
> We hope that our explanations can help clarify our novelty and answer the question on the influence of subgraph sizes. We are also more than happy to address any remaining concern.

---

> > ### Comment · Reviewer_cucL · 2023-11-19
> > **thanks for the clarification**
> >
> > thank you for the answers to my questions. My questions have been clarified. I'll keep my rating as weak accept.

---

> > > ### Author Response · Authors · 2023-11-20
> > > **Thank you for your support towards our work**
> > >
> > > We are happy that the questions from the reviewer have been clarified successfully. We sincerely appreciate the support from the reviewer!

---

### Official Review · Reviewer_DdYK · 2023-10-31

**Soundness:** 3 good
**Presentation:** 2 fair
**Contribution:** 3 good
**Rating:** 6
**Confidence:** 2

**Summary:**

The paper presents a systematic analysis of the memory explosion problem in continual learning on expanding graphs and introduces the PDGNNs-TEM framework, which effectively reduces memory complexity while incorporating topological information for memory replay.
In scenarios where new nodes and edges are continually added to a graph, traditional models may experience catastrophic forgetting when adapting to new data while maintaining performance on old data. Memory replay techniques, successful in other domains, are applied to this problem, but they lead to a memory explosion issue due to the necessity of storing topological neighborhood structures.
To mitigate this problem, the PDGNNs-TEM framework reduces the memory space by decoupling trainable parameters from individual nodes/edges.

**Strengths:**

1.  The authors have provided a comprehensive and in-depth discussion of the relevant literature, encompassing various domains such as continual learning and graph neural networks.

2. The authors conduct comprehensive experiments on multiple datasets, evaluating their method against a range of baselines. The results show that PDGNNs-TEM outperforms existing techniques, achieving better average accuracy with less memory consumption.

**Weaknesses:**

1. A potential weak point of the paper is the perceived limitation in technical novelty. The paper could be interpreted as an application of decoupled GNNs to the continual learning domain, which has already been acknowledged for its computational efficiency benefits. However,  I believe that combining existing, well-established techniques that work efficiently is a valid approach in research. It's essential to leverage the best tools available to address complex problems effectively. In this case, applying decoupled GNNs to the continual learning context is a sensible choice, as it capitalizes on the advantages of both areas. However, it would be beneficial to describe clearly the paper's contribution, impact, and novel techniques.

2. Reproducibility and Open Access: It is not mentioned whether the authors plan to make their code, dataset, or pre-trained models available for the research community. This could be a weak point, as open access to these resources is essential for the reproducibility and widespread adoption of the proposed approach.

**Questions:**

See above.

---

> ### Author Response · Authors · 2023-11-15
> **Responses to Reviewer DdYK (Part1)**
>
> We sincerely appreciate the reviewer for recognizing our in-depth discussion on the literature, as well as the strong and comprehensive empirical results. We have carefully prepared responses to the concerns/questions from the reviewer below.
>
> **Q1. A potential weak point of the paper is the perceived limitation in technical novelty. The paper could be interpreted as an application of decoupled GNNs to the continual learning domain, which has already been acknowledged for its computational efficiency benefits. However, I believe that combining existing, well-established techniques that work efficiently is a valid approach in research. It's essential to leverage the best tools available to address complex problems effectively. In this case, applying decoupled GNNs to the continual learning context is a sensible choice, as it capitalizes on the advantages of both areas. However, it would be beneficial to describe clearly the paper's contribution, impact, and novel techniques.**
>
> **A1.** Thanks for the nice suggestion. We are happy to further clarify our contribution, novelty, and impact in the following.
>
> **1. A novel and effective framework.** As recognized by the reviewer, our proposed PDGNNs-TEM framework, for the first time, resolves the memory explosion problem and enable full preservation of the topological information, which significantly improves the performance on multiple large datasets, especially in the challenging class-IL scenario.
>
> Besides, the contribution of PDGNNs (backbone part of PDGNNs-TEM) lies in presenting a general framework, in which TEs can be derived. Some existing decoupled GNNs can be viewed as instantiations of PDGNNs, but not all of them. As mentioned by the reviewer, the existing decoupled GNNs focus on improving the computational efficiency. Accordingly, they focus on decoupling the neighborhood aggregation and the node feature transformation, without the specific consideration on decoupling the trainable parameters, e.g. APPNP [1], PTA [2], etc. In these decoupled GNNs, the trainable parameters are still entangled with individual nodes in the computation ego-subgraph, therefore TEs cannot be derived and our TEM cannot be implemented. In contrast, our PDGNNs framework explicitly requires decoupling the trainable parameters from the computation ego-subgraphs, and the TEs can be derived (**Section 3.3**).
>
>
> **2. Theoretical contribution.** In this work, we made the first theoretical attempt to analyze the relationship between the stored the data and the remaining ones in the field of continual learning on expanding graphs. This effort resulted in the theoretical results on the pseudo-training effect (**Section 3.5**).
>
> **3. A novel sampling technique (**Section 3.6**).** Based on the theoretical findings, we proposed the coverage maximization sampling technique. This technique could enlarge the coverage ratio and further boost the performance, especially when the memory budget is tight (**Table 1, page 7**).
>
> **4. Impact.** Our proposed PDGNNs-TEM can generally benefit various graph data related research areas. As introduced in the **Introduction (paragraph 1)**, since a majority of real-world graphs are expanding over time with distribution shift, **the studied problem is practically important in many real-world applications,** and are attracting researchers from various communities.
>
> For example, in the recommender system research, new products and new users are continuously added to the purchasing network. In this case, the recommender system have to keep learning new patterns emerging in the new data without forgetting the patterns learnt from previous observations [4,5]. Besides, knowledge graph is also growing over time with new knowledge added, and a machine learning model is required to keep adapting to new knowledge without forgetting the previous ones [3]. Moreover, expanding graphs also exist in many other fields, like papers are constantly increasing in citation networks, new users/posts constantly emerge in social networks, etc. In all these applications, the core problem is the continual learning on expanding graphs studied in our paper, and are extensively studied by researchers [6,7,8,9,10].
>
> Our proposed method is a general solution for representation learning in this continual learning scenario, and can be applied to all these different applications. This is also demonstrated by our comprehensive experiments on different graph types including social network, product co-purchasing network, and citation networks.

---

> ### Author Response · Authors · 2023-11-15
> **Responses to Reviewer DdYK (Part2)**
>
> [1] Klicpera, Johannes, Aleksandar Bojchevski, and Stephan Günnemann. "Predict then propagate: Graph neural networks meet personalized pagerank." arXiv preprint arXiv:1810.05997 (2018).
>
> [2] Dong, Hande, et al. "On the equivalence of decoupled graph convolution network and label propagation." Proceedings of the Web Conference 2021. 2021.
>
> [3] Ahrabian, Kian, et al. "Structure aware experience replay for incremental learning in graph-based recommender systems." Proceedings of the 30th ACM International Conference on Information & Knowledge Management. 2021.
>
> [4] Xu, Yishi, et al. "Graphsail: Graph structure aware incremental learning for recommender systems." Proceedings of the 29th ACM International Conference on Information & Knowledge Management. 2020.
>
> [5] Daruna, Angel, et al. "Continual learning of knowledge graph embeddings." IEEE Robotics and Automation Letters 6.2 (2021): 1128-1135.
>
> [6] Carta, Antonio, et al. "Catastrophic forgetting in deep graph networks: an introductory benchmark for graph classification." arXiv preprint arXiv:2103.11750 (2021).
>
> [7] Zhang, Xikun, Dongjin Song, and Dacheng Tao. "Cglb: Benchmark tasks for continual graph learning." Advances in Neural Information Processing Systems 35 (2022): 13006-13021.
>
> [8] Liu, Huihui, Yiding Yang, and Xinchao Wang. "Overcoming catastrophic forgetting in graph neural networks." Proceedings of the AAAI conference on artificial intelligence. Vol. 35. No. 10. 2021.
>
> [9] Galke, Lukas, et al. "Lifelong learning of graph neural networks for open-world node classification." 2021 International Joint Conference on Neural Networks (IJCNN). IEEE, 2021.
>
> [10] Han, Yi, Shanika Karunasekera, and Christopher Leckie. "Graph neural networks with continual learning for fake news detection from social media." arXiv preprint arXiv:2007.03316 (2020).

---

> ### Author Response · Authors · 2023-11-15
> **Responses to Reviewer DdYK (Part3)**
>
> **Q2. Reproducibility and Open Access: It is not mentioned whether the authors plan to make their code, dataset, or pre-trained models available for the research community. This could be a weak point, as open access to these resources is essential for the reproducibility and widespread adoption of the proposed approach.**
>
> **A2.**
> Thanks for the nice concern.
> We will make all of the code, dataset, and the pre-trained models be publicly available in the final version.
>
> We sincerely hope that our responses can address the concerns of the reviewer. We are also happy to address any remaining concern.

---

### Official Review · Reviewer_1cA5 · 2023-11-01

**Soundness:** 3 good
**Presentation:** 2 fair
**Contribution:** 2 fair
**Rating:** 3
**Confidence:** 3

**Summary:**

This paper presents a general framework, Parameter Decoupled Graph Neural Networks (PDGNNs) with Topology-aware
Embedding Memory (TEM) to tackle the memory explosion problem for continually expanding graphs.
The main contribution is the reduction of memory space complexity and improved model performance.

**Strengths:**

- The paper presents a good theoretical analysis of the PDGNN training
- The memory usage reduction of the proposed method is significant

**Weaknesses:**

- The PDGNN and TEM definitions are vague. I hope the paper can include concrete algorithms to showcase how they are implemented. - - Overall, the formation of PDGNN looks overly complex to me. I hope the authors can showcase the true insights behind PDGNN, potentially via an ablation study, demonstrating which component really works.
- The experimental datasets are limited. First, it is not clear why Cora and OGB datasets are proper for the expanding graphs problem. I hope the authors can evaluate the methods on more practically useful expanding graph datasets.
- The main claim, memory cost reduction, is not validated. There is no memory analysis in the main paper. I found the discussions on the Appendix, but I think it should appear in the main paper and be more thoroughly discussed.
- Overall, the scope of the problem has limited value. There are not many practical use cases with this setting.

**Questions:**

It is confusing for readers to understand the relationships between PDGNN and TEM. What makes a PDGNN? Does it mean we combine any MPNN with TEM?

---

> ### Author Response · Authors · 2023-11-15
> **Responses to Reviewer 1cA5 (Part1)**
>
> We sincerely appreciate the reviewer for recognizing our theoretical merits and our contribution in significantly reducing the memory complexity. We have carefully responded to every concern or question of the reviewer. We have also seriously taken the suggestions and revised the paper. The revised parts are highlighted in blue.
>
> **Q1. The PDGNN and TEM definitions are vague. I hope the paper can include concrete algorithms to showcase how they are implemented. Overall, the formation of PDGNN looks overly complex to me.**
>
> **A1.**
> Thanks for this nice suggestion. To further clarify the definitions of PDGNNs and TEM, we have **added detailed algorithms** in **Appendix C.3 (page 29)** to explain the workflow of PDGNNs-TEM step by step, which are also provided below.
>
>
> ___________
> **Algorithm** Training of PDGNNs-TEM
> _____________
> **Input:** Training task sequence $\mathcal{S}$ = {$\mathcal{G}\_{1}$, $\mathcal{G}\_{2}$,...,$\mathcal{G}\_{T}$}, function $\mathrm{f}\_{topo} (\cdot)$, $\mathrm{f}\_{out}(\cdot; \boldsymbol{\theta})$, sampling function $\mathrm{sampler}(\cdot, \cdot)$, memory budget $n$, loss function $l(\cdot,\cdot)$, neural network optimizer $\mathrm{Optimizer(\cdot,\cdot)}$, number of epochs $M$.
>
> **Output:** Constructed memory buffer $\mathcal{TEM}$, trained parameters $\boldsymbol{\theta}$.
>
> &nbsp;&nbsp;1: Initialize $\mathcal{TEM}$ = {}.
>
> &nbsp;&nbsp;2: **for each** $\mathcal{G} \in \mathcal{S}$ do
>
> &nbsp;&nbsp;3: &nbsp;&nbsp;&nbsp;&nbsp; **for each** $\mathrm{epoch} \in$ {$1,...,M$} do
>
> &nbsp;&nbsp;4: &nbsp;&nbsp;&nbsp;&nbsp;&nbsp;&nbsp;&nbsp;&nbsp; Calculate the TEs for nodes in node set $\mathbb{N}\_{\tau}$ of $\mathcal{G}\_{\tau}$.
>
> &nbsp;&nbsp;5: &nbsp;&nbsp;&nbsp;&nbsp;&nbsp;&nbsp;&nbsp;&nbsp; {$\mathbf{e}\_v \mid \mathbf{e}\_v = \mathrm{f}\_{topo}(\mathcal{G}\_{v}^{sub}), v \in \mathbb{V}\_{\tau}$}
>
> &nbsp;&nbsp;6: &nbsp;&nbsp;&nbsp;&nbsp;&nbsp;&nbsp;&nbsp;&nbsp; Calculate final prediction of nodes in $\mathbb{N}\_{\tau}$
>
> &nbsp;&nbsp;7: &nbsp;&nbsp;&nbsp;&nbsp;&nbsp;&nbsp;&nbsp;&nbsp; {$\hat{\mathbf{y}}\_v \mid \hat{\mathbf{y}}\_v=\mathrm{f}\_{out}(\mathbf{e}\_v;\boldsymbol{\theta}) , v \in \mathbb{V}\_{\tau}$}
>
> &nbsp;&nbsp;8: &nbsp;&nbsp;&nbsp;&nbsp;&nbsp;&nbsp;&nbsp;&nbsp; Calculate loss of the current task $\tau$
>
> &nbsp;&nbsp;9: &nbsp;&nbsp;&nbsp;&nbsp;&nbsp;&nbsp;&nbsp;&nbsp; $\mathcal{L}\_{\tau} = \sum\_{v \in \mathbb{V}\_{\tau}} l(\mathrm{f}\_{out}(\mathbf{e}\_v; \boldsymbol{\theta}), \mathbf{y}\_v) $
>
> 10: &nbsp;&nbsp;&nbsp;&nbsp;&nbsp;&nbsp;&nbsp;&nbsp; Calculate the TEs for nodes in memory $\mathcal{TEM}$
>
> 11: &nbsp;&nbsp;&nbsp;&nbsp;&nbsp;&nbsp;&nbsp;&nbsp; {$\mathbf{e}\_v \mid \mathbf{e}\_v = \mathrm{f}\_{topo}(\mathcal{G}\_{v}^{sub}), v \in \mathcal{TEM}$}
>
> 12: &nbsp;&nbsp;&nbsp;&nbsp;&nbsp;&nbsp;&nbsp;&nbsp; Calculate final prediction of nodes in $\mathcal{TEM}$
>
> 13: &nbsp;&nbsp;&nbsp;&nbsp;&nbsp;&nbsp;&nbsp;&nbsp; {$\hat{\mathbf{y}}\_v \mid \hat{\mathbf{y}}\_v=\mathrm{f}\_{out}(\mathbf{e}\_v;\boldsymbol{\theta}) , v \in \mathbb{V}\_{\tau}$}
>
> 14: &nbsp;&nbsp;&nbsp;&nbsp;&nbsp;&nbsp;&nbsp;&nbsp; Calculate the auxiliary loss for preventing forgetting
>
> 15: &nbsp;&nbsp;&nbsp;&nbsp;&nbsp;&nbsp;&nbsp;&nbsp; $\mathcal{L}\_{aux} = \sum\_{\mathbf{e}\_w \in \mathcal{TEM}} l(\mathrm{f}\_{out}(\mathbf{e}\_w;\boldsymbol{\theta}), \mathbf{y}\_w) $
>
> 16: &nbsp;&nbsp;&nbsp;&nbsp;&nbsp;&nbsp;&nbsp;&nbsp; Calculate balancing parameter $\lambda$ according to the class sizes in $\mathbb{N}\_{\tau}$ and $\mathcal{TEM}$
>
> 17: &nbsp;&nbsp;&nbsp;&nbsp;&nbsp;&nbsp;&nbsp;&nbsp; Calculate the total loss
>
> 18: &nbsp;&nbsp;&nbsp;&nbsp;&nbsp;&nbsp;&nbsp;&nbsp; $\mathcal{L}\_{total} = \mathcal{L}\_{\tau} + \lambda \mathcal{L}\_{aux}$
>
> 19: &nbsp;&nbsp;&nbsp;&nbsp;&nbsp;&nbsp;&nbsp;&nbsp; Calculating the gradients of the trainable parameters $\frac{\partial \mathcal{L}\_{total}}{\partial \boldsymbol{\theta}}$.
>
> 20: &nbsp;&nbsp;&nbsp;&nbsp;&nbsp;&nbsp;&nbsp;&nbsp; Update $\boldsymbol{\theta}$ with the optimizer
>
> 21: &nbsp;&nbsp;&nbsp;&nbsp;&nbsp;&nbsp;&nbsp;&nbsp; $\boldsymbol{\theta} = \mathrm{Optimizer}(\boldsymbol{\theta},\frac{\partial \mathcal{L}\_{total}}{\partial \boldsymbol{\theta}})$
>
> 22: &nbsp;&nbsp;&nbsp;&nbsp;  **end for each**
>
> 23: &nbsp;&nbsp;&nbsp;&nbsp; Update the memory buffer
>
> 24: &nbsp;&nbsp;&nbsp;&nbsp; $ \mathcal{TEM}=\mathcal{TEM} \; \bigcup \; \mathrm{sampler}(${$\mathbf{e}\_v \mid v\in \mathbb{V}\_{\tau}$}, $n)$
>
> 25: **end for each**

---

> ### Author Response · Authors · 2023-11-15
> **Responses to Reviewer 1cA5 (Part2)**
>
> _____________________
> **Algorithm** Inference of PDGNNs-TEM
> _______________________
> **Input:** Test task sequence $\mathcal{S}$ = {$\mathcal{G}\_{1}$, $\mathcal{G}\_{2}$,...,$\mathcal{G}\_{T}$}, function $\mathrm{f}\_{topo} (\cdot)$, $\mathrm{f}\_{out}(\cdot; \boldsymbol{\theta})$.
>
> **Output:** Constructed memory buffer $\mathcal{TEM}$, trained parameters $\boldsymbol{\theta}$.
>
> &nbsp;&nbsp;1: **for each** $\mathcal{G} \in \mathcal{S}$
>
> &nbsp;&nbsp;2: &nbsp;&nbsp;&nbsp;&nbsp;Calculate the TEs for nodes in node set $\mathbb{N}\_{\tau}$ of $\mathcal{G}\_{\tau}$
>
> &nbsp;&nbsp;3: &nbsp;&nbsp;&nbsp;&nbsp;{$\mathbf{e}\_v \mid \mathbf{e}\_v = \mathrm{f}\_{topo}(\mathcal{G}\_{v}^{sub}), v \in \mathbb{V}\_{\tau}$}
>
> &nbsp;&nbsp;4: &nbsp;&nbsp;&nbsp;&nbsp;Calculate final prediction of nodes in $\mathbb{N}\_{\tau}$
>
> &nbsp;&nbsp;5: &nbsp;&nbsp;&nbsp;&nbsp;{$\hat{\mathbf{y}}\_v \mid \hat{\mathbf{y}}\_v=\mathrm{f}\_{out}(\mathbf{e}\_v;\boldsymbol{\theta}) , v \in \mathbb{V}\_{\tau}$}
>
> &nbsp;&nbsp;6: **end for each**
>
> During training, when learning on each task, function $\mathrm{f}\_{topo}(\cdot)$ will first generate the TE for each node, which is subsequently fed into the output function $\mathrm{f}\_{out}(\cdot; \boldsymbol{\theta})$ to get the final prediction. After obtaining the predictions, the loss $\mathcal{L}\_{\tau}$ is calculated based on the predictions and the ground-truth labels. The same procedure is also conducted for data in the memory buffer $\mathcal{TEM}$ to get the loss $\mathcal{L}\_{aux}$. Then a total loss is obtained by summing up $\mathcal{L}\_{\tau}$ and $\mathcal{L}\_{aux}$ with $\lambda$ to balance the contribution from classes with different sizes. After learning each task, the memory buffer will be updated with new representative data selected by the sampling function $\mathrm{sampler}(\cdot, \cdot)$.
>
> During testing, only the inference stage is conducted. The given data will sequentially go through function $\mathrm{f}\_{topo}(\cdot)$ and $\mathrm{f}\_{out}(\cdot;\boldsymbol{\theta})$ to get the final prediction.

---

> ### Author Response · Authors · 2023-11-15
> **Responses to Reviewer 1cA5 (Part3)**
>
> **Q2. I hope the authors can showcase the true insights behind PDGNN, potentially via an ablation study, demonstrating which component really works.**
>
> **A2.** Thanks for this nice concern on the functionality of each component. We are more than happy to have the opportunity to further clarify the insights.
>
> Briefly speaking, the insights are:
>
> 1. Topological-aware Embedding Memory (TEM) is the main reason of the significant performance improvement.
>
> 2. The functionality of the PDGNNs framework is to enable the implementation of TEM, i.e. TEM cannot work without PDGNNs.
>
> 3. Coverage maximization sampling serves to further boost the performance especially when the memory budget is tight.
>
> First, our proposed PDGNNs-TEM mainly consists of the following components: 1. the memory buffer TEM with the sampling strategy, 2. the backbone model PDGNNs consisting of $f_{topo}$ and $f_{out}$.
>
> To investigate the functionality of each component, typical ablation study would try to remove one of the component and check how the performance is influenced. Within our PDGNNs-TEM, the backbone PDGNNs cannot be removed, and only TEM is removable. We could definitely provide the results of PDGNNs without TEM, as shown in the table below. Just like any GNN without continual learning technique, PDGNNs alone will suffer from catastrophic forgetting. This is empirically justified in the paper by the results of 'Fine-tune' shown in **Table 2 and 3 (page 7 and 9)**.
>
> | C.L.T.  | CoraFull | | OGB-Arxiv  | | Reddit  | | OGB-Products  | |
> | ---------| ------- |-|-------|-|------- |-|-------|-|
> | |AP/\% $\uparrow$ | AF/\% $\uparrow$ | AP/\% $\uparrow$ | AF/\% $\uparrow$ |AP/\% $\uparrow$ | AF/\% $\uparrow$ |AP/\% $\uparrow$ | AF/\% $\uparrow$ |
> | PDGNNs (without TEM) | 7.3 $\pm$ 0.8|- 93.3$\pm$4.5 |7.4$\pm$1.5|-77.2$\pm$2.8| 6.6$\pm$0.7 |-95.4$\pm$7.2 | 10.3$\pm$1.8 |-85.4$\pm$3.3|
> | PDGNNs (with TEM) |81.9$\pm$0.1|-3.9$\pm$0.1 | 53.2$\pm$0.2|-14.7$\pm$0.2|96.6$\pm$0.0|-2.6$\pm$0.1 | 73.9$\pm$0.1|-10.9$\pm$0.2|
>
> However, for our continual learning research, this only justifies the existence of catastrophic forgetting phenomenon, but cannot reflect why our continual learning technique is superior over the baselines.
>
> Therefore, in the paper, we have designed experiments to investigate our PDGNNs-TEM from multiple perspectives, and to demonstrate the functionality of each component.
>
> First, the advantage of our TEM is that it completely store the topological information. Therefore, to test **whether full topological information preservation is crucial**, we include two other memory replay based baselines ER-GNN and SSM in **Table 2 and 3 (page 7 and 9)**. ER-GNN stores individual nodes and completely ignore the topological information, while SSM stores sparsified computation ego-subgraphs with partial topological information preserved. According to the results, our PDGNNs-TEM outperforms SSM, and SSM outperforms ER-GNN. In other words, the more topological information is preserved, the better the performance is. This demonstrates the importance of topological, and justifies that **TEM is a crucial part**.
>
> Second, as introduced in **Section 3.2 and 3.3**, Topology-aware Embeddings (TEs) can only be derived in the proposed PDGNNs framework. Therefore, to implement TEM, which is the key to the performance improvement, PDGNNs is an indispensable component.
>
> Finally, as introduced in **Section 3.5 and 3.6**, with the PDGNNs framework, we theoretically justified the existence of pseudo-training effect, based on which we develop the coverage maximization sampling technique. This sampling technique can select the TEs that are more effective for alleviating the forgetting problem, which is advantageous especially for the tight memory budget situation, as empirically justified in **Section 4.3**.
>
> We hope the explanations above could clarify the insights of our work.

---

> ### Author Response · Authors · 2023-11-15
> **Responses to Reviewer 1cA5 (Part4)**
>
> **Q3. The experimental datasets are limited. First, it is not clear why Cora and OGB datasets are proper for the expanding graphs problem. I hope the authors can evaluate the methods on more practically useful expanding graph datasets.**
>
> **A3.** Thanks for this suggestion. As introduced in **Section 4.1**, the datasets are actually public benchmark datasets in the field, and are specially pre-processed for continual learning on expanding graphs.
>
> Specifically, CoraFull and OGB datasets are pre-processed by the Continual Graph Learning Benchmark (CGLB) [1]. In CGLB, the datasets are first split into subgraphs (tasks) according to the classes. Then the subgraphs are gradually connected to simulate the expanding process of the graph. Moreover, these datasets are representative for the naturally expanding graphs. First, CoraFull and OGB-Arxiv are constructed based on real-world citation networks, and new papers with potential new classes are constantly emerging in real-world citation networks. Second, the OGB-Products is a co-purchasing network constructed based on real-world shopping data from Amazon. On real-world shopping platforms like Amazon, new products and users also emerge continuously.
>
> These benchmark datasets, as well as the splitting scheme, are widely adopted by the field of continual learning on expanding graphs [2,3]. This class based dataset splitting can effectively ensure the diversity in task distributions, therefore is adopted in different continual learning research on expanding graphs [1,4], images [5,6], etc.
>
>
> [1] Zhang, Xikun, Dongjin Song, and Dacheng Tao. "Cglb: Benchmark tasks for continual graph learning." Advances in Neural Information Processing Systems 35 (2022): 13006-13021.
>
> [2] Sun, Li, et al. "Self-supervised continual graph learning in adaptive riemannian spaces." Proceedings of the AAAI Conference on Artificial Intelligence. Vol. 37. No. 4. 2023.
>
> [3] Liu, Huihui, Yiding Yang, and Xinchao Wang. "Overcoming catastrophic forgetting in graph neural networks." Proceedings of the AAAI conference on artificial intelligence. Vol. 35. No. 10. 2021.
>
> [4] Carta, Antonio, et al. "Catastrophic forgetting in deep graph networks: an introductory benchmark for graph classification." arXiv preprint arXiv:2103.11750 (2021).
>
> [5] Rebuffi, Sylvestre-Alvise, et al. "icarl: Incremental classifier and representation learning." Proceedings of the IEEE conference on Computer Vision and Pattern Recognition. 2017.
>
> [6] Van de Ven, Gido M., and Andreas S. Tolias. "Three scenarios for continual learning." arXiv preprint arXiv:1904.07734 (2019).
>
>
> **Q4. The main claim, memory cost reduction, is not validated. There is no memory analysis in the main paper. I found the discussions on the Appendix, but I think it should appear in the main paper and be more thoroughly discussed.**
>
> **A4.** Thanks for this nice suggestion. In the revised version of the main paper, we have carefully added the discussion on the memory cost reduction in a new subsection (**Section 4.5**) to discuss the empirical memory efficiency PDGNNs-TEM.
>
> In Section 4.5, the actual memory usage (the number of float32 values) of different memory based methods after learning each entire dataset are provided in **Table 4 ,page 9**. As a reference, the memory consumption of storing full computation ego-subgraph is also calculated. The table is also provided below for convenience.
>
> | C.L.T.  | CoraFull | OGB-Arxiv  | Reddit | OGB-Products |
> | ---------| ------- |-|-------|-|
> | Full Subgraph | 7,264M | 35M | 2,184,957M | 5,341M |
> | ---------| ------- |-|-------|-|
> | GEM |7,840M | 86M| 329M| 82M|
> | ER-GNN |61M |2M |12M |3M |
> | SSM | 732M| 41M|193M |37M |
> | PDGNNs-TEM |37M | 2M| 9M|2M |

---

> ### Author Response · Authors · 2023-11-15
> **Responses to Reviewer 1cA5 (Part5)**
>
> **Q5. Overall, the scope of the problem has limited value. There are not many practical use cases with this setting.**
>
> **A5.** Thanks for this concern.
>
> As introduced in the **Introduction (paragraph 1)** of the paper, since a majority of real-world graphs are expanding over time with distribution shift, **the studied problem is practically important in many real-world applications,** and are **attracting researchers from various communities**.
>
> For example, in the recommender system research, new products and new users are continuously added to the purchasing network. In this case, the recommender system have to keep learning new patterns emerging in the new data without forgetting the patterns learnt from previous observations [1,2]. Besides, knowledge graph is also growing over time with new knowledge added, and a machine learning model is required to keep adapting to new knowledge without forgetting the previous ones [3]. Moreover, expanding graphs also exist in many other fields, like papers are constantly increasing in citation networks, new users/posts constantly emerge in social networks, etc. In all these applications, the core problem is the continual learning on expanding graphs studied in our paper, and are extensively studied by researchers [4,5,6,7,8].
>
> Our proposed method is a general solution for representation learning in this continual learning scenario, and can be applied to all these different applications. This is also demonstrated by our comprehensive experiments on different graph types including social network, product co-purchasing network, and citation networks.
>
> [1] Ahrabian, Kian, et al. "Structure aware experience replay for incremental learning in graph-based recommender systems." Proceedings of the 30th ACM International Conference on Information & Knowledge Management. 2021.
>
> [2] Xu, Yishi, et al. "Graphsail: Graph structure aware incremental learning for recommender systems." Proceedings of the 29th ACM International Conference on Information & Knowledge Management. 2020.
>
> [3] Daruna, Angel, et al. "Continual learning of knowledge graph embeddings." IEEE Robotics and Automation Letters 6.2 (2021): 1128-1135.
>
> [4] Carta, Antonio, et al. "Catastrophic forgetting in deep graph networks: an introductory benchmark for graph classification." arXiv preprint arXiv:2103.11750 (2021).
>
> [5] Zhang, Xikun, Dongjin Song, and Dacheng Tao. "Cglb: Benchmark tasks for continual graph learning." Advances in Neural Information Processing Systems 35 (2022): 13006-13021.
>
> [6] Liu, Huihui, Yiding Yang, and Xinchao Wang. "Overcoming catastrophic forgetting in graph neural networks." Proceedings of the AAAI conference on artificial intelligence. Vol. 35. No. 10. 2021.
>
> [7] Galke, Lukas, et al. "Lifelong learning of graph neural networks for open-world node classification." 2021 International Joint Conference on Neural Networks (IJCNN). IEEE, 2021.
>
> [8] Han, Yi, Shanika Karunasekera, and Christopher Leckie. "Graph neural networks with continual learning for fake news detection from social media." arXiv preprint arXiv:2007.03316 (2020).
>
>
>
>
> **Q6. It is confusing for readers to understand the relationships between PDGNN and TEM. What makes a PDGNN? Does it mean we combine any MPNN with TEM?**
>
> **A6.** Thanks for this nice question. We are happy to further clarify *what makes a PDGNN*, as well as *the relation between PDGNN and TEM*.
>
> First, as analyzed in **Section 3.2 and 3.3**, the principle of designing PDGNNs is to decouple the trainable parameters from the neighborhood aggregation operation (for capturing topological information). In this way, the trainable parameters act on the computation ego-subgraph as a whole (via the TEMs), and are not involved in computation with individual nodes in the computation ego-subgraph, thereby the storage of the complete computation ego-subgraph is avoided. To summarize, **PDGNN is a framework of GNNs in which the trainable parameters do not participate in computation with individual nodes in the computation ego-subgraph, but not any MPNN**.
>
> Second, as explained in **the paragraph above Equation (6)** of the paper, TEM refers to the memory buffer that stores Topology-aware embeddings (TEs). And **TE is the connection between TEM and PDGNNs.** Specifically, as introduced in **Section 3.3**, only within the PDGNNs framework can we derive the TE as a single-vector representation of the computation ego-subgraph. Therefore, **TEM can only be implemented with the PDGNNs framework**. For an arbitrary MPNN that does not decouple the trainable parameters, TE cannot be derived, therefore TEM cannot be used either.
>
> To summarize, **PDGNNs are specially designed for obtaining TEs and using TEM**. Also, TEM can only be used with PDGNNs, but **not any MPNN**.
>
> We sincerely hope that our explanations can address the concerns and answer the questions from the reviewer. We are also more than happy to address any remaining concern.

---

> ### Author Response · Authors · 2023-11-18
> **A friendly reminder on the approaching discussion deadline**
>
> We sincerely thank the reviewer for the constructive comments.
>
> In our detailed responses, we have tried our best to address all the concerns.
>
> Given that the discussion period is approaching to the end, we will appreciate if the reviewer can advise whether there is any remaining concern, and we are more than happy to address them.

---

> ### Author Response · Authors · 2023-11-20
> **A friendly reminder that the discussion is ending in 2 days**
>
> We sincerely thank the reviewer for the constructive comments.
>
> Given that the discussion period is ending in 2 days, we will appreciate if the reviewer can advise whether there is any remaining concern, and we are more than happy to address them.

---

> ### Author Response · Authors · 2023-11-22
> **A friendly reminder that the discussion is ending in 1 day**
>
> We sincerely thank the reviewer for the constructive comments.
>
> Given that the discussion period is ending in 1 day, we will appreciate if the reviewer can advise whether there is any remaining concern, and we are more than happy to address them.

---

### Official Review · Reviewer_jbPM · 2023-11-01

**Soundness:** 3 good
**Presentation:** 3 good
**Contribution:** 3 good
**Rating:** 6
**Confidence:** 2

**Summary:**

This paper proposes a new framework for continual learning of GNNs. Methods such as memory replay solve continuous learning issues such as catastrophic forgetting, but there are major problems from a memory perspective. The proposed method does not store all subgraphs, but instead stores vectors equivalent to key nodes in a topology-aware manner and utilizes them for training. This has been effective in reducing the amount of memory and the challenge of continuous learning.

**Strengths:**

A new framework that greatly improves the memory problem in continuous learning on expanding graphs and has high performance is proposed, and experiments have shown its effectiveness.

**Weaknesses:**

Basically, this paper is well written, but there is one thing that bothers me. This paper describes the avoidance of memory explosion as an issue, and specifically argues for a reduction from O(nd^L) to O(n). The reduction in memory order is almost trivial, and is also discussed in a simplified manner in the text. On the other hand, no experiments with actual memory perspectives have been conducted. Memory reduction may have contributed to the improved performance shown in the experiment, but not directly. For example, a comparison of the possible scale of response between ER-GNNs and PDGNNs should be mentioned in a clear manner.

**Questions:**

In addition to the WEAKNESS content, there are questions about the following:

I would like to know the impact of TEM sampler settings, I understand that coverage maximization sampling is recommended, but are there other sampling methods with catastrophic results? Does the sample size of coverage maximization sampling make a critical difference? If possible, experimental results would be useful for actual use.

---

> ### Author Response · Authors · 2023-11-15
> **Responses to Reviewer jbPM**
>
> We sincerely appreciate the reviewer for recognizing the novelty and the empirical effectiveness of our proposed framework. We have carefully prepared responses to the concerns of the reviewer. We have also seriously taken the suggestions and made revisions to the paper. The revised parts are highlighted in blue.
>
> **Q1. This paper describes the avoidance of memory explosion as an issue, and specifically argues for a reduction from $O(nd^L)$ to O(n). The reduction in memory order is almost trivial, and is also discussed in a simplified manner in the text. On the other hand, no experiments with actual memory perspectives have been conducted. Memory reduction may have contributed to the improved performance shown in the experiment, but not directly. For example, a comparison of the possible scale of response between ER-GNNs and PDGNNs should be mentioned in a clear manner.**
>
> **A1.**
> Thanks for this nice concern on the memory reduction. Originally, the discussion on the memory reduction was included in **Appendix B.2**. We have now added a new section (**Section 4.5**) in the paper to discuss this issue. In Section 4.5, the actual memory usage (the number of float32 values) of different memory based methods after learning each entire dataset, including ER-GNN mentioned by the reviewer, are provided in **Table 4 ,page 9**. As a reference, the memory consumption of storing full computation ego-subgraph is also calculated. The table is also provided below for convenience.
>
> | C.L.T.  | CoraFull | OGB-Arxiv  | Reddit | OGB-Products |
> | ---------| ------- |-|-------|-|
> | Full Subgraph | 7,264M | 35M | 2,184,957M | 5,341M |
> | ---------| ------- |-|-------|-|
> | GEM |7,840M | 86M| 329M| 82M|
> | ER-GNN |61M |2M |12M |3M |
> | SSM | 732M| 41M|193M |37M |
> | PDGNNs-TEM |37M | 2M| 9M|2M |
>
>
> **Q2. I would like to know the impact of TEM sampler settings, I understand that coverage maximization sampling is recommended, but are there other sampling methods with catastrophic results? Does the sample size of coverage maximization sampling make a critical difference? If possible, experimental results would be useful for actual use.**
>
> **A2.** Thanks for this nice question on the sampling strategy, which is an important contribution in our work. We have actually provided experimental results (**Table 1, page 7**) regarding this issue and discussed about it (**Section 4.3**), as explained below. The table is also provided below for convenience
>
> || Ratio of dataset / \% | 0.02 | 0.1 | 1.0  | 5.0 | 40.0 |
> | ---- |----| ---- |----|----|----|----|
> AA/% | Uniform sampling |12.0$\pm$1.1|24.1$\pm$1.7|42.2$\pm$0.3|50.4$\pm$0.4|53.3$\pm$0.4 |
> || Mean of feature | 12.6$\pm$0.1|25.3$\pm$0.3|42.8$\pm$0.3|50.4$\pm$0.7|53.3$\pm$0.2 |
> || Coverage Maximization |**14.9$\pm$0.8**|**26.8$\pm$1.8**|**43.7$\pm$0.5**|**50.5$\pm$0.4**|**53.4$\pm$0.1**|
> Cov. ratio/% | Uniform samp. |0.1$\pm$0.1|0.3$\pm$0.0|3.5$\pm$0.9|15.9$\pm$1.1|84.8$\pm$1.5 |
> ||Mean of feature  |0.2$\pm$0.4|0.6$\pm$0.3|7.1$\pm$0.6|29.6$\pm$1.7|91.1$\pm$0.1 |
> ||Coverage Maximization|**0.5$\pm$1.1**|**2.9$\pm$1.8**|**22.5$\pm$1.6**|**46.3$\pm$0.6**|**92.8$\pm$0.0**|
>
>
> In this table, we show the performance of coverage maximization as well as two other samplers, uniform sampling and Mean of Feature (MoF). Results with different memory budget (sample size) are also shown in Table 1. According to the result, PDGNNs-TEM can avoid catastrophic forgetting with different samplers. When the budget is tight (sample size is small), the advantage of Coverage Maximization sampling is more significant.
>
> Above all, we summarize our answer to the two questions as:
>
> First, our key design to avoid catastrophic forgetting is the full preservation of the topological information in TEM. Therefore, **PDGNNs-TEM can perform well with different TEM samplers**.
>
> Second, **the sample size does make a critical difference**, just like any memory replay based method in continual learning. The more data is preserved from the previous tasks, the less forgetting is suffered from.

---

> > ### Comment · Reviewer_jbPM · 2023-11-18
> > **Thank you for your answers**
> >
> > Thank you for your answers to my questions. My questions have been clarified. I hope that the paper will be made better by polishing. I recognize that the proposed method is still in the pilot study stage and should be improved upon. Since there is no 7 rating as a system for this time, the score itself will be maintained, but I would like to add that I have a positive impression of it.

---

> > > ### Author Response · Authors · 2023-11-19
> > > **Thank you for your positive feedback and support towards our work**
> > >
> > > We are more than happy to see that the questions of the reviewer have been clarified.
> > > We sincerely appreciate the reviewer's positive feedback and the support to our work!

---

### Official Review · Reviewer_wMQc · 2023-11-03

**Soundness:** 3 good
**Presentation:** 4 excellent
**Contribution:** 2 fair
**Rating:** 6
**Confidence:** 3

**Summary:**

The paper proposes a strategy for reducing the memory complexity from $O(nd^L)$ to $O(n)$ for continual learning (CL) on expanding
graphs. More specifically, the strategy removes the requirement of storing the entirety of ego-subgraphs which is normally needed for replay buffers. The framework, Parameter Decoupled Graph Neural Networks (PDGNNs) with Topology-aware
Embedding Memory (TEM), is competitive against several state-of-the-art baselines over various CL tasks. In addition to empirical evidence of efficacy, the authors contribute theory as well.

**Strengths:**

1. Addresses an important issue: memory complexity of GNN training is a major bottleneck. The proposed approach introduces a simple, yet effective technique for reducing the memory complexity while avoiding catastrophic forgetting.

2. The empirical evidence is strong. A wide variety of baselines are proposed and the authors' method uses significantly less memory in the replay buffer while outperforming most baselines across several tasks. Large datasets are also included to support the strategy.

**Weaknesses:**

1. The authors do not suggest a novel $f_{topo}$ (the embedding function) which limits the novelty of the work. Essentially, the authors are proposing to reduce the memory complexity of ego-subgraph topological information using known techniques.

2. Some issues with theory: (a) I believe Definition 1 is both too strong and not rigorous enough, “if optimizing $\theta$ with $G_v^{sub}$ or $e_v$ are equivalent,” does this mean the same stationary points are achieved? Additionally, since $e_v$ is an embedding, this would be a highly unusual possibility as any type of compression leads to different solution spaces. (b) Theorem 1:
“It is equivalent to training PDGNNs with each node w in $G_v^{sub}$ with $G_v^{sub}$ being a pseudo computation ego-subgraph and $y_v$ being a pseudo label,” in my opinion, this is a near circular outcome. You have defined the embedding as $e_v$ as being a representation of $G_v^{sub}$ such that optimization is equivalent, which is strong enough to allow all these results to naturally fall out.

Minor:
Misformatted citations. Some examples: Section 2.2: misformatted citation Kipf & Welling (2016)
Section 3.2: “prevent forgetting LopezPaz & Ranzato (2017); Rebuffi et al. (2017).”

“Take the Reddit dataset (Hamilton et al., 2017) as a concrete example, its average degree is 492, and even with a 2 layer MPNN, the buffer size will be easily intractable.” This sentence was repeated in the intro.

**Questions:**

Please clarify my concerns regarding theory. In particular, the existence of a non-trivial topology-aware embedding as in Definition 1, as I think the definition refers to the existence of a very strong function.

---

> ### Author Response · Authors · 2023-11-15
> **Responses to Reviewer wMQc (Part1)**
>
> We sincerely appreciate the reviewer for recognizing that our proposed framework effectively addresses the important memory explosion issue, with strong empirical results on large datasets and theoretical merits. The concerns and questions are very constructive to further improve our work, and we have carefully prepared responses below. We have also seriously made necessary revisions to the paper, which are highlighted in blue.
>
> **Q1. The authors do not suggest a novel $f_{topo}$ (the embedding function) which limits the novelty of the work. Essentially, the authors are proposing to reduce the memory complexity of ego-subgraph topological information using known techniques.**
>
> **A1.** Thanks for this nice concern, and we are happy to have the opportunity to further clarify it.
>
> *1. A new framework:* as the reviewer noticed, $f_{topo}$ is one component of our proposed PDGNNs framework and it is formulated based on existing techniques. The entire framework, which consists of both PDGNNs and TEM, is the first in the field that can fully leverage the topological information with highly efficiently memory usage. As recognized by the reviewer and other reviewers, this new framework can significantly boost the performance on large datasets.
>
> *2. Innovative theoretical analysis (**Section 3.5**):* In the field of continual learning on expanding graphs, we provide the first attempt to theoretically analyze the relationship between the sampled and the remaining nodes.
>
> *3. A novel sampling technique with theoretical foundations (**Section 3.6**):* Based on the theoretical analysis, we developed the Coverage Maximization Sampling, which is also empirically justified to be effective especially when the memory budget is tight.
>
> All of these have not been accomplished by existing works and are important issues for continual learning on expanding graphs.

---

> ### Author Response · Authors · 2023-11-15
> **Responses to Reviewer wMQc (Part2)**
>
> **Q2. Some issues with theory: (a) I believe Definition 1 is both too strong and not rigorous enough, “if optimizing $\mathbf{\theta}$ with $\mathcal{G}^{sub}_v$ or $\mathbf{e}_v$ are equivalent,” does this mean the same stationary points are achieved? Additionally, since $\mathbf{e}_v$ is an embedding, this would be a highly unusual possibility as any type of compression leads to different solution spaces. (b) Theorem 1: “It is equivalent to training PDGNNs with each node w in $\mathcal{G}^{sub}_v$ with $\mathcal{G}^{sub}_v$ being a pseudo computation ego-subgraph and $\mathbf{y}_v$ being a pseudo label,” in my opinion, this is a near circular outcome. You have defined the embedding as $\mathbf{e}_v$ as being a representation of $\mathcal{G}^{sub}_v$ such that optimization is equivalent, which is strong enough to allow all these results to naturally fall out.**
>
> **A2.(a)** Thanks for these nice concerns on the theory.
>
> First of all, we agree with the reviewer that 'any type of compression leads to different solution spaces'. For example, in our PDGNNs framework, different choices of $f_{topo}$ would result in different embedding spaces for the TEs. However, as stated in **Definition 1**, the optimization equivalence is based on a 'given GNN parameterized with $\mathbf{\theta}$'. In other words, the TEs are not generated once and universally fit all GNNs. Instead, different qualified (capable of generating TEs) GNNs generate TEs in different embedding spaces, and the equivalence is only required within the specified GNN. In our PDGNNs framework, the function $f_{topo}$ is formulated with different choices in the experiments (**Section 3.4**). Different choices of $f_{topo}$ lead to different embedding spaces of TEs, and the optimization equivalence is only required after $f_{topo}$ is specified and the embedding space is fixed.
>
> After clarification on the above issue, we can address the two concerns in Q2.(a) from the reviewer.
>
> 1. *As for 'since $\mathbf{e}_v$ is an embedding, this would be a highly unusual possibility as any type of compression leads to different solution spaces.'* Yes, the embeddings (TEs) are model dependent, and each qualified GNN generates their own TEs. The TEs derived within one specific GNN may not full-fill the 'optimization equivalence' requirement of Definition 1 for another GNN.
>
> 2. *As for 'if optimizing $\mathbf{\theta}$ with $\mathcal{G}^{sub}_v$ or $\mathbf{e}_v$ are equivalent, does this mean the same stationary points are achieved?'* Yes, equivalence in terms of optimization requires achieving the same stationary points. For example, in our PDGNNs framework, we provided proof in Appendix A.1 to show that optimizing with TEs and computation ego-subgraphs results in same gradient on the trainable parameters $\mathbf{\theta}$ with respect to the loss function, at each optimization step. Therefore, after optimization, the same stationary points can be achieved.
>
> Based on the reviewer's constructive comments, we have carefully rewritten Definition 1 to strengthen that the *optimization equivalence is only required with respect to each specified GNN structure*, which is highlighted in blue in the updated version (**page 4**).
>
> **A2.(b)** Thanks for this nice concern. The reviewer's observation (Definition 1 is the foundation of Theorem 1) is correct. However, Definition 1 alone is insufficient for developing Theorem 1 for the following reasons.
>
> 1. Definition 1 describes a desired surrogate (TE) of the computation ego-subgraph. But it does not guarantee the existence of such a surrogate or specify the model structure for deriving TEs. Differently, Theorem 1 is built upon the constructed PDGNNs framework, with detailed analysis on the pseudo-training effect within this specific framework. For example, Theorem 1 describes the pseudo-training in both situations where $\mathrm{f}_{out}$ is linear or not, as well as how exactly the contribution of the nodes are re-scaled in both situations.
>
> 2. The optimization equivalence in Definition 1 and the pseudo-training in Theorem 1 are actually different. The optimization equivalence in Definition 1 is between the TE and the computation ego-subgraph of a specific node $v$, instead of every node $w$ within the subgraph, while Theorem 1 describes the relationship between training with $v$ and its neighbors in its computation ego-subgraph $\mathcal{G}_v^{sub}$.
>
> 3. Moreover, Theorem 1 leads to our novel coverage maximization sampling technique, which cannot be directly derived from Definition 1.

---

> ### Author Response · Authors · 2023-11-15
> **Responses to Reviewer wMQc (Part3)**
>
> **Q3. Minor: Misformatted citations. Some examples: Section 2.2: misformatted citation Kipf and Welling (2016) Section 3.2: “prevent forgetting LopezPaz and Ranzato (2017); Rebuffi et al. (2017).” “Take the Reddit dataset (Hamilton et al., 2017) as a concrete example, its average degree is 492, and even with a 2 layer MPNN, the buffer size will be easily intractable.” This sentence was repeated in the intro.**
>
> **A3.** Thanks for pointing out these issues. We have carefully corrected every issue in the revised version. Specifically, the misformatted citations are updated into the correct format, and the last paragraph of Section 3.2 containing a duplicate sentence from Introduction is re-written and highlighted in blue.
>
> **Q4. Please clarify my concerns regarding theory. In particular, the existence of a non-trivial topology-aware embedding as in Definition 1, as I think the definition refers to the existence of a very strong function.**
>
> **A3.** Thanks for further clarifying this question. As explained in A2.(a), the reviewer is correct that it is hard to guarantee the existence of a non-trivial topology aware embedding universally for different GNNs, which is not implied by our Definition 1. Instead, Definition 1 requires the optimization equivalence to exist with respect to each specific GNN that is capable of generating TEs. We have carefully rewritten **Definition 1** and highlighted it in blue (**page 4**).
>
> We hope our explanations can help in the clarification of the concerns. We are also happy to address any remaining concerns.

---

### Author Response · Authors · 2023-11-22
**A summary of the discussion period**

Dear Area Chairs and Reviewers,

Again, we sincerely thank all reviewers for the recognition of our contribution and the valuable comments, which are very constructive to further improve our work. To further facilitate the reviewing process, we briefly summarize the reviews and our responses below. We would really appreciate it if the reviewers could let us know if the concerns are resolved.

1. Reviewer wMQc is mainly concerned with details of our theoretical results and the novelty of one module in our framework. We have provided detailed responses to further clarify our theoretical analysis, as well as clarification on our contribution and novelty.

2. Reviewer jbPM has two questions/concerns, which are already resolved by our rebuttal

3. Reviewer 1cA5 has 5 main concerns. First, regarding the suggestion of providing algorithms to better showcase the implementation of our model, we have provided two detailed algorithms, which explain every step of our method.
Second, regarding the concern about the true insights of our method, we have provided detailed explanations of how the experimental results demonstrate the effectiveness of each part of our model.
Third, regarding the concern on why the datasets adopted are proper for our task, we have explained that the datasets are widely adopted benchmark datasets for our studied problem. We also provided detailed explanations of the rationale of these datasets.
Fourth, regarding the suggestion to strengthen memory reduction in the main paper, we have carefully added a new section with experimental results in the main paper to discuss this issue.
Fifth, regarding the concern that the studied problem has limited practical use, we have provided various examples with references to published papers to justify the practical importance of the problem studied in our work.

4. Reviewer DdYK has two suggestions/concerns. Regarding the suggestion to further clarify our contribution and novelty, we provided detailed responses to explain our novelty and contribution from different perspectives. Second, regarding on the concern on reproducibility, we explained that we would release all the code and datasets in the final version

5. Reviewer cucL has two concerns, which are successfully resolved by our responses.

---

### Meta-Review · Area_Chair_1aUn · 2023-12-09

**Metareview:**

Paper systematically studies memory explosion problem in continual learning on expanding graphs. Memory replay is required to prevent catastrophic forgetting but authors claim this leads to memory explosion due to the need for storing ego-neighborhood topology of each node. Authors tackle this issue by PDGNN-TEM framework which creates topology-aware non-trainable features (TEM) for each node. They are instantiated as as linear combination of input features of nodes in the L-hop neighborhood of that node. Then TEM is passed through an MLP to create the prediction. Since TEM’s are non-trainable they can be sampled and stored in memory buffers. Authors study an important problem. Sampling technique seems very interesting and useful in practice.

However, there some unaddressed concerns such as:
1. Not sure if we can call PGDNN a graph neural network as there is no graph convolution operator. PDGNN is more like a regular NN which uses graph-level features. Different from this, in the decoupled GNN of (Dong et al., 2021) the propagation layer comes after feature transformation, but that would not work in this setting.

2. Theorem 1: “Pseudo-training” is not well-defined, theorem/proof isn’t rigorous and it is not clear if its implications are meaningful. Authors may be able to further bring out their intuition by adding further assumptions on the relation between neighboring nodes. Rescaled TEM embedding is a totally a different embedding that the true TEM embedding for the neighborhood node.

3. “Equation (8) is also highly efficient especially for large graphs due to the absence of iterative neighborhood aggregations.”: computing powers of adjacency matrix are equivalent to iterative aggregation, only difference is that it can be done with scalar 1 dimension instead of embedding dimension in standard message passing in GNNs.

4. “For a fair comparison, all methods including fout(·; θ) of PDGNNs are set as 2-layer with 256 hidden dimensions,”: It is not clear if this is fair, as it is understood that all competing methods including finetune and joint training can employ complex and deeper GNNs. For example, currently Joint training performs very similar to PGDNN-TEM which is counter-intuitive because joint training can use actual GNNs. Currently the manuscript reads like they are not doing so. Further no code is provided to verify this. Along the same manner, in the memory consumption table, it would be nice to add finetuning and Joint baselines as well.

  Further authors note that PGDNN sometimes even outperforms Joint training which is considered to be an upperbound. This is hard to believe since joint training has more information than PDGNN. This points to potential unfair choice of baseline, or it may be missing regularization techniques or hyperparameter tuning. Authors could further experimentally explore this discrepancy.

5. Complexity analysis is not provided but it is stated that there is a reduction from O(nd^L) to O(n). But shouldn’t the original complexity should be upper bounded by total number of nodes in a batch/task as: O(max(nd^L, max_r |V_r|))? And if n = 0.4 max_r |V_r| as taken in the best performing choice in Table 1, there is no improvement in complexity. Still there could be some practical benefits due to subsampling, but memory consumption given in Table 4 seems very different from the above calculation and it may be counting the same node in multiple ego-graphs.

**Justification For Why Not Higher Score:**

There are a few concerns as noted in meta-review

**Justification For Why Not Lower Score:**

N/A

---

### Decision · Program_Chairs · 2024-01-16

Reject